# Label Delay in Online Continual Learning

Botos Csaba[1,3,*],          Wenxuan Zhang[2,*],          Matthias Müller[3],          Ser-Nam Lim[4],
Mohamed Elhoseiny[2],          Philip H.S. Torr[1],          Adel Bibi[1]

[1]University of Oxford
[2]King Abdullah University of Science and Technology
[3]Intel Labs
[4]University of Central Florida

## Abstract

A critical yet often overlooked aspect in online continual learning is the label delay, where new data may not be labeled due to slow and costly annotation processes. We introduce a new continual learning framework with explicit modeling of the label delay between data and label streams over time steps. In each step, the framework reveals both unlabeled data from the current time step $t$ and labels delayed with $d$ steps, from the time step $t - d$. In our extensive experiments amounting to 25000 GPU hours, we show that merely increasing the computational resources is insufficient to tackle this challenge. Our findings highlight significant performance declines when solely relying on labeled data when the label delay becomes significant. More surprisingly, state-of-the-art Self-Supervised Learning and Test-Time Adaptation techniques that utilize the newer, unlabeled data, fail to surpass the performance of a naïve method that simply trains on the delayed supervised stream. To this end, we propose a simple, robust method, called Importance Weighted Memory Sampling that can effectively bridge the accuracy gap caused by label delay by prioritising memory samples that resemble the most to the newest unlabeled samples. We show experimentally that our method is the least affected by the label delay factor, and successfully recovers the accuracy of the non-delayed counterpart. The implementation for reproducing our experiments can be found at https://github.com/botcs/label-delay-exp.

## 1 Introduction

Machine learning models have become the de facto standard for a wide range of applications, including social media [1], finance [2], and healthcare [3]. However, these models usually struggle when the distribution from which the data is sampled is constantly changing over time, which is common in real-world scenarios. This challenge continues to be an active area of research known as Continual Learning (CL). However, most prior art in CL examines this problem with a presumption of the immediate availability of labels once the data is collected. This assumption rarely holds in real-world scenarios.

Consider the task of monitoring recovery trends in patients after surgeries. Doctors gather health data from numerous post-operative patients regularly. However, this data does not immediately indicate broader recovery trends or potential common complications. To make informed determinations, several weeks of extensive checks and tests across multiple patients are needed. Only after these checks are completed, the gathered data can be labeled as indicating broader "recovery" or "complication" trends. However, by the time the data is gathered, assessed, labeled, and a model is trained, new

---

*Equal Contribution

38th Conference on Neural Information Processing Systems (NeurIPS 2024).

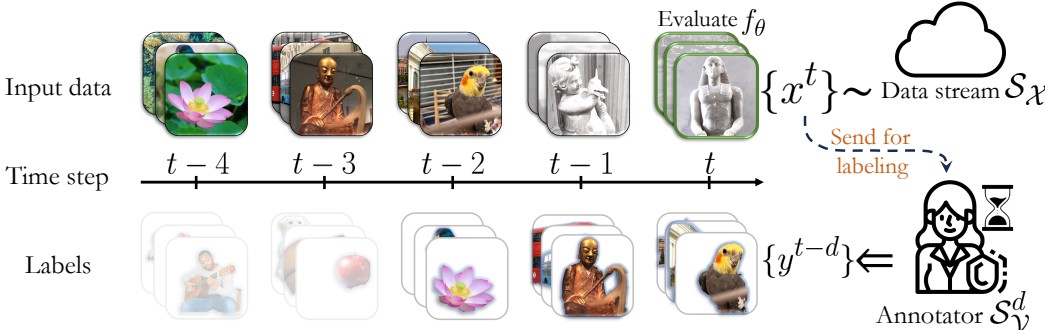

Figure 1: **Illustration of label delay.** This figure shows a typical Continual Learning (CL) setup with label delay due to annotation. At every time step $t$, the data stream $\mathcal{S}_\mathcal{X}$ reveals a batch of unlabeled data $\{x^t\}$, on which the model $f_\theta$ is evaluated (highlighted with green borders). The data is then sent to the annotator $\mathcal{S}_\mathcal{Y}$ who takes $d$ time steps to provide the corresponding labels. Consequently, at time step $t$ the batch of labels $\{y^{t-d}\}$ corresponding to the input data from $d$ time steps before becomes available. The CL model can be trained using the **delayed labeled data** (shown in color) and the **newest unlabeled data** (shown in grayscale). In this example, the stream reveals three samples at each time step and the annotation delay is $d = 2$.

patient data might follow trends that do not exist in the training data yet. This leads to a repeating cycle: collecting data from various patients, assessing the trends, labeling the data, training the model, and then deploying it on new patients. The longer this cycle takes, the more likely it is going to affect the model's reliability, a challenge we refer to as **label delay**.

In this paper, we propose a CL setting that explicitly accounts for the delay between the arrival of new data and the corresponding labels, illustrated by Figure 1. In our proposed setting, the model is trained continually over discrete time steps with a label delay of $d$ steps. At each step, two batches of data are revealed to the model: unlabeled new samples from the current time step $t$, and the labels of the samples revealed at the step $t - d$. First, we show the naïve approach where the model is only trained with the labeled data while ignoring all unlabeled data. While this forms a strong baseline, its performance suffers significantly from increasing the delay $d$. We find that simply increasing the number of parameter updates per time step does not resolve the problem. Hence, we examine a number of popular approaches which incorporate the unlabeled data to improve this naïve baseline. We investigate semi-supervised learning, self-supervised learning and test-time adaptation approaches which are motivated for slightly different but largely similar settings. Surprisingly, out of 12 different methods considered, none could outperform the naïve baseline given the same computational budget. Motivated by our extensive empirical analysis of prior art in this new setting, we propose a simple and efficient method that outperforms every other approach across large-scale datasets; in some scenarios it even closes the accuracy gap caused by the label delay. Our contributions are threefold:

- We propose a new formal Continual Learning setting that factors label delay between the arrival of new data and the corresponding labels due to the latency of the annotation process.

- We conduct extensive experiments ($\sim 25,000$ GPU hours) on various Online Continual Learning datasets, such as CLOC [4], CGLM [5], FMoW [6] and Yearbook [7]. Following recent prior art on Budgeted Continual Learning [8, 9], we compare the best performing Self-Supervised Learning [10], Semi-Supervised Learning [11] and Test Time Adaptation [12] methods and find that none of them outperforms the naïve baseline that simply ignores the label delay and trains a model on the delayed labeled stream.

- We propose **I**mportance **W**eighted **M**emory **S**ampling to rehearse past labeled data most similar to the most recent unlabeled data, bridging the gap in performance. IWMS outperforms the naïve method significantly and improves over Semi-Supervised, Self-Supervised Learning and Test-Time Adaptation methods across diverse delay and computational budget scenarios with a negligible increase in computational complexity. We further present an in-depth analysis of the proposed method.

---
**Algorithm 1** Single OCL time step with Label Delay
---
1. The Stream $\mathcal{S}_{\mathcal{X}}$ reveals a batch of images $\{x_i^t\}_{i=1}^n \sim \mathcal{D}_t$;
2. The model $f_{\theta_t}$ makes predictions $\{\hat{y}_i^t\}_{i=1}^n$ for the new revealed batch $\{x_i^t\}_{i=1}^n$;
3. The Annotator $\mathcal{S}_{\mathcal{Y}}^d$ reveals labels $\{y_i^{t-d}\}_{i=1}^n$;
4. The model $f_{\theta_t}$ is evaluated by comparing the predictions $\{\hat{y}_i^t\}_{i=1}^n$ and true labels $\{y_i^t\}_{i=1}^n$, where the true labels are only for testing;
5. The model $f_{\theta_t}$ is updated to $f_{\theta_{t+1}}$ using labeled data $\cup_{\tau=1}^{t-d}\{(x_i^\tau, y_i^\tau)\}_{i=1}^n$ and unlabeled data $\cup_{\tau=t-d}^{t}\{x_i^\tau\}_{i=1}^n$ under a computational budget $\mathcal{C}$.
---

## 2   Related Work

**Label Delay in Online Learning.** While the problem of delayed feedback has been studied in the online learning literature [13, 14], the scope is limited to problems of spam detection and other synthetically generated, low-complexity data [15, 11] and often views input images as "side info"[16]. Additionally, methods and error bounds proposed in[17, 18, 19, 20] are more focused on expert selection rather than representation learning, most of which cannot generalize to unstructured, large-scale image classification datasets.

**Continual Learning.** Early work on continual learning primarily revolved around task-based continual learning [21, 22], while recent work focuses on the task-free continual learning setting[23, 24, 4]. This scenario poses a challenge for models to adapt as explicit task boundaries are absent, and data distributions evolve over time. GDumb[25] and BudgetCL[8] demonstrate that minimalistic methods can outperform most offline and online continual learning approaches. RealtimeOCL [9] shows that Experience Replay [26] is the most effective method, outperforming more popular continual learning methods, such as ACE [21], LwF [27], RWalk [28], PoLRS [4], MIR [29] and GSS [30], when methods are normalized by their computational complexities.

**Semi-Supervised Learning.** While the labels arrive delayed, our setting allows the models to use new unlabeled data immediately. Possible directions to leverage the most recent unlabeled data entail Pseudo-Labeling (or often referred to as their broader category, Semi-Supervised Learning) methods [11] and Self-Supervised Semi-Supervised Learning (S4L) methods [31]. Pseudo-labeling techniques predict the labels of the samples before their true labels become available to estimate the current state of the joint distribution of input and output pairs. This in turn allows the model to fit its parameters on the estimated data distribution. On the other hand, S4L integrates self-supervised learning, such as predicting the rotation of an image or the relative location of image patches, with the semi-supervised learning framework. We replace the early self-supervised tasks of S4L [31] with more recent objectives from Balestriero et al. [10].

**Test-Time Adaptation.** Besides semi-supervised learning, TTA methods are also designed to adapt models with unlabeled data, sampled from a similar distribution as the evaluation samples. Entropy regularization methods like SHOT [32] and TENT [33] update the feature extractor or learnable parameters of the batch-normalization layers [34] to minimize the entropy of the predictions. SAR [35] incorporates an active sampling scheme to filter samples with noisy gradients. More recent works consider Test Time Adaptation in an online setting [36] or Continual Learning setting [37]. In our experiments, we fine-tune the model with ER [26] across time steps and adapt a copy of the model with TTA to the most recent input samples at each time step.

## 3   Problem Formulation

We follow the conventional online continual learning problem definition proposed by Cai et al. [4]. In such a setting, we seek to learn a model $f_\theta : \mathcal{X} \to \mathcal{Y}$ on a stream $\mathcal{S}$ where for each time step $t \in \{1, 2, \dots\}$ the stream $\mathcal{S}$ reveals data from a time-varying distribution $\mathcal{D}_t$ sequentially in batches of size $n$. At every time step, $f_\theta$ is required to predict the labels of the coming batch $\{x_i^t\}_{i=1}^n$ first. Followed by this, the corresponding labels $\{y_i^t\}_{i=1}^n$ are immediately revealed by the stream. Finally, the model is updated using the most recent training data $\{(x_i^t, y_i^t)\}_{i=1}^n$.

This setting, however, assumes that the annotation process is instantaneous, i.e., the time it takes to provide the ground truth for the input samples is negligible. In practice, this assumption rarely holds. It is often the case that the rate at which data is revealed from the stream $\mathcal{S}$ is faster than the rate at

---

**Algorithm 2** Importance Weighted Memory Sampling

---
1. At time step $t$, for each unsupervised batch of size $n$, $\{x_i^t\}_{i=1}^n$, the model $f_\theta$ computes predictions $\{\widetilde{y}_i^t\}_{i=1}^n$;
2. For every predicted label $\widetilde{y}_i^t$, select labeled samples from the memory buffer $\{(x_j^M, y_j^M)\}$ where $y_j^M = \widetilde{y}_i^t$;
3. Compute pairwise cosine feature similarities $K_{i,j} = \cos\left(h(x_i^t), h(x_i^M)\right)$ between each unlabeled sample $x_i^t$ and selected memory samples $x_j^M$;
4. Select the most relevant supervised samples $(x_k^M, y_k^M)$ by sampling $k \in \{1 \ldots |M|\}$ from a multinomial distribution with parameters $K_{i,:}$;
5. Update the model $f_\theta$ using the selected supervised samples, aiming to match the distribution of the unlabeled data.

---

which labels for the unlabeled data can be collected, as opposed to it being instantaneously revealed. To account for this delay in accumulating the labels, we propose a setting that accommodates this lag in label availability while still allowing for the model to be updated with the most recent unlabeled data. We modify the previous setting in which labels of the data revealed at time step $t$ will only be revealed after $d$ time steps in the future.

At every time step $t$, the Annotator $\mathcal{S}_{\mathcal{Y}}^d$ reveals the labels for the samples from $d$ time steps before, i.e., $\{(x_i^{t-d}, y_i^{t-d})\}_{i=1}^n$, while the data stream $\mathcal{S}_{\mathcal{X}}$ reveals data from the the current time step, i.e., $\{x_i^t\}_{i=1}^n$. Recent prior art [25, 8, 9] introduces more reasonable and realistic comparisons between continual learning methods by imposing a computational complexity constraint on the methods. Similarly to [25, 8, 9], in our experiments the models are given a fixed computational budget $\mathcal{C}$ to update the model parameters from $\theta_t$ to $\theta_{t+1}$ for every time step $t$. To that end, our new proposed setting can be formalized per time step $t$, alternatively to the classical OCL setting, as described in Algorithm 1.

Note that this means at each time step $t$, the stream reveals a batch of *non-corresponding* images $\{x_i^t\}_{i=1}^n$ and labels $\{y_i^{t-d}\}_{i=1}^n$, as illustrated in Figure 1. With the label delay of $d$ time steps, the images themselves revealed from time step $t - d$ to time step $t$ can be used for training, despite that labels are not available.

A *naïve* way to solve this problem is to discard the unlabeled images and only train on labeled data $\cup_{\tau=1}^{t-d}\{(x_i^\tau, y_i^\tau)\}_{i=1}^n$, However, it worth noting that the model is still evaluated on the most recent samples from $\mathcal{S}_{\mathcal{X}}$. Thus, training on the labeled training data leads to the model at least being $d$ steps delayed. Since in our setting the distribution from which the training and evaluation samples are drawn from is not stationary, this discrepancy severely hinders the performance, as discussed in detail in Section 5.

Furthermore, we shall show in Section 6 that the existing paradigms, such as Test-Time Adaptation and Semi-Supervised Learning, struggle to effectively utilise newer, unlabeled data to bridge the aforementioned discrepancy. Our observations indicate that the primary failure is from the excessive computational demands of processing unlabeled data. To that end, we propose Importance Weighted Memory Sampling that prioritises performing gradient steps on labeled samples that resemble the most recent unlabeled samples.

## 4 IWMS: Importance Weighted Memory Sampling

To mitigate the challenges posed by label delay in online continual learning, we introduce a novel method named **Importance Weighted Memory Sampling (IWMS)**. Recognizing the limitation of traditional approaches that either discard unlabeled data or utilize it in computationally expensive ways, IWMS aims to bridge the gap between the current distribution of unlabeled data and the historical distribution of labeled data. Instead of directly adapting the model to fit the newest distribution with unlabeled data, which is inefficient due to the lack of corresponding labels, IWMS cleverly adjusts the sampling process from a memory buffer. This method ensures that the distribution of selected samples closely matches the distribution of the most recent unlabeled batch. This nuanced selection strategy allows the continual learning model to effectively adapt to the most recent data trends, despite the delay in label availability, by leveraging the rich information embedded in the memory buffer.

As discussed in Section 5, using the most recent labeled samples for training leads to over-fitting the model to an outdated distribution. Thus, we replace the newest supervised data by a batch which

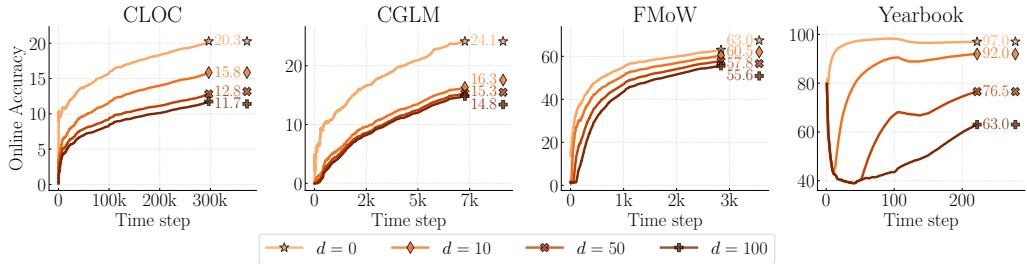

Figure 2: **Effects of Varying Label Delay.** The performance of a *Naïve* Online Continual Learner model gradually degrades with increasing values of delay $d$.

we sample from the memory buffer, such that the distribution of the selected samples matches the newest unlabeled data distribution. The sampling process is detailed in Algorithm 2. It consists of two stages: first, at each time step $t$, for every unsupervised sample $x_i^t$ in the batch of size $n$, we compute the prediction $\widetilde{y}_i^t$, and select every labeled sample from the memory buffer $(x_j^M, y_j^M)$ such that the true label of the selected samples matches the predicted label $y_j^M = \widetilde{y}_i^t$. In the second stage, we compute the pairwise cosine feature similarities $\mathbf{K}_{i,j}$ between the unlabeled sample $x_i^t$ and the selected memory samples $x_j^M$ by $\mathbf{K}_{i,j} = \cos\left(h(x_i^t), h(x_j^M)\right)$, where $h$ represents the learned feature extractor part of $f_\theta$, directly before the final classification layer. Finally, we select the most relevant supervised samples $(x_k^M, y_k^M)$ by sampling $k \in \{1 \ldots |M|\}$ from a multinomial distribution with parameters $\mathbf{K}_{:,j}$. Thus, we rehearse samples from the memory which (1) share the same true labels as the predicted labels of the unlabeled samples, (2) have high feature similarity with the unlabeled samples.

To avoid re-computing the feature representation $h(x^M)$ for each sample in the memory buffer after every parameter update, we store the corresponding features of the input data computed for the predictions during the evaluation (Step 4 in Algorithm 1). This technique greatly reduces the computational cost of our method, but comes at the price of using outdated features. Such trade-off is studied in detail by contemporary Self-Supervised Literature [38, 39, 40] observing no significant impact on performance. We ablate the alternative option of selecting samples based only on their similarity in the Supplementary Material A.11.

## 5 The Cost of Ignoring Label Delay

To better understand how label delay influences the performance of a model, we begin with the **Naïve** approach, i.e., ignoring the most recent data points until their label becomes available and exclusively training on outdated labeled samples. More specifically, we are interested in measuring the performance degradation under various label delay $d$ and computational budget $\mathcal{C}$ scenarios. To this end, we conduct experiments over 4 datasets, in 4 computational budget and 3 label delay settings. We analyse the results under normalised computational budget (Section 5.2) and demonstrate that the accuracy drop can be only partially recovered by increasing the computational budget (Section A.5).

### 5.1 Experimental Setup

**Datasets.** We conduct our experiments on four large-scale online continual learning datasets, Continual Localization (CLOC) [4], Continual Google Landmarks (CGLM) [5], Functional Map of the World (FMoW) [6], and Yearbook [7]. The last two are adapted from the Wild-Time challenge [41]. More statistics of the benchmarks are in Supplementary. We follow the same *training* and *validation* set split of CLOC as in [4] and o CGLM as in [5] and the official released splits for FMoW [6] and Yearbook [7].

**Architecture and Optimization.** Similarly to prior work [9, 8], we use ResNet18 [42] for backbone architecture. Furthermore, in our experiments, the stream reveals a mini-batch, with the size of $n = 128$ for CLOC, FMoW, Yearbook and $n = 64$ for CGLM. We use SGD with the learning rate of

0.005, momentum of 0.9, and weight decay of $10^{-5}$. We apply random cropping and resizing to the images, such that the resulting input has a resolution of $224 \times 224$.

**Baseline Method** In our experiments, we refer to the *Naïve* method as the one naively training one labeled data. We apply the *state of the art* continual learning mechanism under computational constrains [9], Experience Replay (ER) [26], to eliminate the need to compare with other continual learning methods . The memory buffer size is consistently $2^{19}$ samples throughout our experiments unless stated otherwise. The First-In-First-Out mechanism [26, 4] to update the buffer. The discussion of the effectiveness of the memory buffer can be found in Section A.18. We report the Online Accuracy [4] at each time step in Step 4 of Algorithm 1 under label delay $d$. In our quantitative comparative analysis, for simplicity, we use the final Online Accuracy scores, denoted by $\mathrm{Acc}_d$. Refer to Section A.17 for the discussion of more metrics.

**Computational Budget and Label Delay.** Normalising the computational budget is necessary for fair comparison across CL methods, thus, we define $\mathcal{C} = 1$ as the number of FLOPs required to make one backward pass with a ResNet18 [42], similarly to BudgetCL[8] and RealtimeOCL[9]. We discuss its relation to the wall-clock training time in Section A.16. When performing experiments with a larger computational budget, we take integer multiplies of $\mathcal{C}$ to apply $\mathcal{C}$ parameter update steps per stream time steps. The proposed label delay factor $d$ represents the amount of time steps the labels are delayed with. Note that, for $\mathcal{C} = 1, d = 0$, our experimental setting is identical to prior art[4, 9].

## 5.2 Observations

In Figure 2, we analyze how varying the label delay $d \in \{0, 10, 50, 100\}$ impacts the performance of Naïve on four different datasets, CLOC [4], CGLM [8], FMoW [6] and Yearbook [7]. The label delay impacts the online accuracy differently across all scenarios, thus, below we provide our observations case-by-case.

On **CLOC**, the non-delayed ($d = 0$) Naïve achieves $\mathrm{Acc}_0 = 20.2\%$, whereas the heavily delayed counterpart ($d = 100$) suffers significantly from the label delay, achieving only $\mathrm{Acc}_{100} = 11.7\%$. Interestingly, label delay influences the accuracy in a monotonous, but non-linear fashion, as half of the accuracy drop is caused by a very small amount of delay: $\mathrm{Acc}_{10} - \mathrm{Acc}_0 = -4.4\%$. In contrast, the accuracy degradation slows down for larger delays,i.e., the accuracy gap between two larger delay scenarios ($d = 50 \rightarrow 100$) is rather marginal $\mathrm{Acc}_{100} - \mathrm{Acc}_{50} = -1.1\%$. We provide further evidence on the monotonous and smooth properties of the impact of label delay with smaller increments of $d$ in the Supplementary Material A.3.

For **CGLM** the accuracy gap landscape looks different: the majority of the accuracy decrease occurs by the smallest delay $d = 0 \rightarrow 10$, resulting in a $\mathrm{Acc}_{10} - \mathrm{Acc}_0 = -7.9\%$ drop. Subsequent increases ($d = 10 \rightarrow 50$ and $d = 50 \rightarrow 100$) impact the performance to a significantly smaller extent: $\mathrm{Acc}_{50} - \mathrm{Acc}_{10} = -1\%$ and $\mathrm{Acc}_{100} - \mathrm{Acc}_{50} = -0.5\%$.

In the case of **FMoW**, where the distribution shift is less imminent (i.e., the data distribution varies less over time), the difference between the delayed and the non-delayed counterparts should be small. This is the case for the satellite image data in the FMoW dataset, where the accuracy drops are $-2.8\%, -2\%, -1.9\%$ for $d = 0 \rightarrow 10 \rightarrow 50 \rightarrow 100$, respectively.

The **Yearbook**'s binary classification experiments highlight an important characteristic: if there is a significant event that massively changes the data distribution, such as the change of men's appearance in the 70's [7] the non-delayed Naïve ($d = 0$) suffers a small drop in Online Accuracy (at the middle of the time horizon $t = 130$), but quickly recovers as more data starts to appear. In contrast, under small and moderate delay ($d = 10, 50$), the decline is more emphasised and the recovery is delayed (at $t = 120, 180$, respectively). Alongside with more detailed investigation, we provide visual examples of the dataset to support our claims in the Supplementary Material A.4.

## 5.3 Section Conclusion

**Over- or Under-fitting.** While in the experiments we report results under a single computational budget $\mathcal{C}$ per dataset, it is reasonable to suspect that the results might look different under smaller or larger budget. To this end, we ablate the effect of $\mathcal{C}$ over various delay scenarios, on multiple datasets in the Supplementary Material A.5.

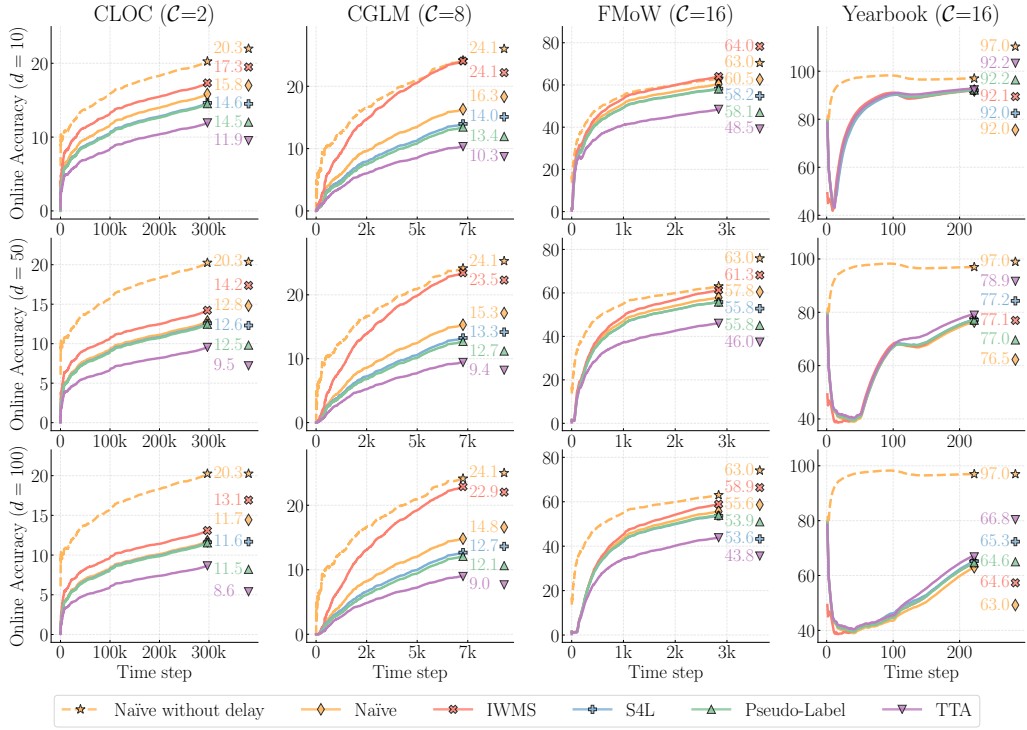

Figure 3: **Comparison of various unsupervised methods.** The accuracy gap caused by the label delay between the *Naïve without delay* and its delayed counterpart *Naïve*. Our proposed method, *IWMS*, consistently outperforms all categories under all delay settings on three out of four datasets.

**Common patterns.** We argue that the consistent, monotonic accuracy degradation, present in all of our experiments, is due to the non-stationary property of the data distribution that creates a distribution shift. Our hypothesis is supported by the findings of Yao et al. [41]. A complementary argument is presented by Hammoud et al. [43], stating that the underlying datasets have high temporal correlations across the labels, i.e., images of the same categories arrive in bursts, allowing an online learning model to easily over-fit the label distribution even without using the input images.

**Motivation for Delay Specific Solutions.** As our experiments suggest so far, label delay is indeed an extremely elusive problem, not only because it inevitably results in an accuracy drop, but because the severity of the drop itself is hard to estimate a-priori. We showed that the accuracy gap always increases monotonically with increasing delay, nevertheless the increase of the gap can be gradual or sudden depending on the dataset and the computation budget. This motivates our efforts of designing special techniques to address the challenges of label delay. In the next set of experiments, we augment the Naïve training by utilizing the input images *before* their corresponding labels become available.

## 6 Utilising Data Prior to Label Arrival

In our proposed label delay experimental setting, we showed the larger the delay the more challenging it is for Naïve, a method that relies only on older labeled data, to effectively classify new samples. This is due to a larger gap in distribution between the samples used for training and for evaluation. This begs the question of whether the new unlabeled data can be used for training to improve over Naïve, as it is much more similar to the data that the model is evaluated on.

We propose four different paradigms for utilizing the unlabeled data, namely, Importance Weighted Memory Sampling (**IWMS**), Semi-Supervised Learning via Pseudo-Labeling (**PL**), Self-Supervised Semi-Supervised Learning (**S4L**) and Test-Time Adaptation (**TTA**). We integrate several methods of each family into our setting and evaluate them under various delays and computational budgets. In particular, we adapt each paradigm individually by augmenting the parameter update (Step 5 of

Algorithm 1) of Naïve, described in detail in the following subsections. Furthermore, to quantify the how much of the accuracy gap ($G_d = \text{Acc}_d^{\text{Naïve}} - \text{Acc}_0^{\text{Naïve}}$) is recovered, we use the formula $R_d^* = \frac{\text{Acc}_d^* - \text{Acc}_d^{\text{Naïve}}}{|G_d|}$, namely the improvement of the method divided by the extent of the accuracy gap for a given delay factor $d$.

## 6.1 Experiment Setup

**Importance Weighted Memory Sampling (IWMS).** The only additional cost of IWMS compared to Naïve is the cost of evaluating the similarity scores, which is still less than $1\%$ of the inference cost for 100K samples, and can be evaluated in parallel, therefore we consider it negligible. Since our method simply replaces the newest supervised samples with the most similar samples from the replay buffer, we do not require any additional backward passes to compute the auxiliary objective. Therefore, the computational budget of our method is identical to the Naïve baseline, i.e.,, $\mathcal{C}_{\text{IMWS}} = 1$.

**Self-Supervised Semi-Supervised Learning.** For integrating S4L methods, we adopt the most effective approach through iterative optimization of both supervised and unsupervised losses. We report the best results across the three main families of contrastive losses, i.e.,, Deep Metric Learning Family (MoCo [38], SimCLR [44],and NNCLR [45]), Self-Distillation (BYOL [46] and SimSIAM [47], and DINO [40]), and Canonical Correlation Analysis (VICReg [48], BarlowTwins [49], SWAV [50], and W-MSE [51]).

For fair comparison, we normalise the computational complexity of the compared methods. According to [8, 9], Naïve augmented with Self-Supervised Learning at each time step takes two backward passes, since they augment each input images to two views, thus $\mathcal{C}_{\text{S4L}} = 2$. We provide further explanation of our S4L adaptation in the Supplementary Material A.2.

**Pseudo-Labeling.** To make use of the newer unlabeled samples, we adopt the most common Semi-Supervised Learning technique [11]: Pseudo-Labeling (PL). To predict the labels of the samples before their true label becomes available we use a surrogate model $g_\phi$. After assigning the predicted labels $\{\tilde{y}_i^t\}$ to each input data $\{x_i^t\}$ at time step $t$ for $i = 1..n$, the main model $f_\theta$ is updated over the union of old, labeled memory samples and new *pseudo-labeled* samples $\{(x_i^\tau, y_i^\tau)\}_{\tau=1}^{t-d} \cup \{(x_i^t, \tilde{y}_i^t)\}$ using standard Cross Entropy loss. Once $f_\theta$ is updated, we update the parameters of the surrogate model $g_\phi$ following the momentum update policy [11] with hyper-parameter $\lambda$, such that $\phi_{\text{new}} = \lambda\phi_{\text{old}} + (1 - \lambda)\theta_{\text{old}}$.

For simplicity, we ignore the computational cost of the surrogate model $g_\phi$ inferring the pseudo-labels $\tilde{y}$. Nevertheless, the main model $f_\theta$ is trained on double the amount of samples as Naïve, $n$ labeled and $n$ pseudo-labeled, therefore we define $\mathcal{C}_{\text{PL}} = 2$.

**Test-Time Adaptation** As done for other paradigms, we have extensively evaluated all reasonable candidates to adapt traditional TTA methods to our setting. We find performing the unsupervised TTA step the most effective when only a single update is taken (in Step 5 of Algorithm 1), exactly before the evaluation step (Step 2 of Algorithm 1) of the next step. Therefore, for all the parameter updates apart from the last one we perform identical steps to Naïve Furthermore, we found TTA updates severely impact the continual learning process of the Naïve when the parameters are iteratively optimised across the two objectives. Thus, before each TTA step, we clone the model parameters $\theta$ to a surrogate model $g_\phi$, by performing the TTA step (with $\epsilon$ hyper-parameter) using the newest batch of unlabeled data $\phi = \theta - \epsilon\nabla_\theta\mathcal{L}_{\text{TTA}}\{x_i^t\}$ and perform the evaluation (Step 2 of Algorithm 1) of the next time step.

To represent the state of the art in TTA, we adapt and compare the following methods: TENT [33], EATA [52], SAR [35], and CoTTA [37], in Figure 3. Furthermore, for the result of our hyper-parameter tuning is provided in the Supplementary Material A.7.

For fair comparison, we train and evaluate all TTA methods under normalised computational budgets (a detailed breakdown of the considerations can be found in the Supplementary Material A.15) More specifically, under a fixed computational budget $\mathcal{C}$, at every time step, we perform $\mathcal{C} - 1$ supervised steps on $f_\theta$ identically to Naïve followed by a single step of TTA.

## 6.2 Observations

Figure 3 illustrates our most important results of our work. It shows to what extent we can recover the accuracy gap caused by the label delay between the *Naïve without delay* and its delayed counterpart *Naïve*. We evaluate our proposed method, *IWMS*, and compare it against the three adopted paradigms, *S4L*, *PL* and *TTA*. We report the best performing method of each paradigm with hyperparameters tuned on the first 10% of each label delay scenario (further detailed in the Supplementary Material A.6 and 12).

**IWMS.** On the largest dataset, containing 39M samples, **CLOC** [4], the accuracy drop of Naïve is $G_d = -4.5\%, -7.5\%, -8.6\%$ for $d = 10, 50, 100$, respectively. Our proposed method, IWMS, achieves $\text{Acc}_d = 17.3\%, 14.2\%, 13.1\%$ final Online Accuracy, which translates to $R_d = 33\%, 19\%, 16\%$ recovery for $d = 10, 50, 100$, respectively. While there is a slow decline over increasing delays, the improvement over Naïve is consistent. On **CGLM** [8], the accuracy drop is $G_d = -7.8\%, -8.8\%, -9.3\%$ for the three increasing delays, respectively. IWMS exhibits outstanding results, $\text{Acc}_d = 24.1\%, 23.5\%, 22.9\%$ meaning that the accuracy gap *is fully recovered* by the method for $d = 10$. More specifically, the recovery is $R_d = 100\%, 93\%, 87\%$ for $d = 10, 50, 100$. The results on **FMoW** [6] are even more surprising, as IWMS not only recovers the accuracy gap but *outperforms* the non-delayed Naïve counterpart in the $d = 10$ scenario. More specifically, the accuracy drops for the increasing delays are $G_d = 2.5\%, 3.2\% 4.4\%$ and $R_d = \underline{140\%}, 67\%, 45\%$. We hypothesise this is due to the fact that under a large $\mathcal{C}$, repeated parameter updates with suboptimal sampling strategies lead to over-fitting to the outdated state of the data distribution, as explained in detail in Section 7. On **Yearbook** [7], IWMS performs on-par with Naïve in every scenario. The accuracy gaps are $G_d = -5\%, -20.5\%, -34\%$ whereas the recover scores are marginal: $R_d = 1\%, 0\%, 0\%$. We argue this is due to two factors: the brevity of the dataset in comparison to the other datasets and the difficulty of the task without prior knowledge on appearance and fashion trends.

**Semi-Supervised Methods.** S4L and PL performs very similarly to each other under all studied scenarios: the largest difference in their performance is $0.7\%$ on Yearbook, under $d = 50$ label delay. Therefore, we report their performance together, picking the better performing variant for numerical comparisons. Notice that in every scenario the delayed Naïve baseline performance is not be achieved, which is due to the computational budget constraint. More specifically, since $\mathcal{C}_{\text{SSL}} = 2 \times \mathcal{C}_{\text{Naïve}}$, optimising the standard classification objective over the older, supervised samples for twice the number of parameter updates is more beneficial across all scenarios than optimising the Pseudo-Labeling classification objective or the Contrastive loss over the newer unlabeled images. In the Supplementary Material 13, we provide further evidence and explanation of this claim. On **CLOC**, S4L slightly outperforms PL by $+0.1\%$ for all label scenarios, however $R_d = -27\%, -2\%, -7\%$ for $d = 10, 50, 100$, respectively. Similarly, on **CGLM**, S4L outperforms PL by $+0.6\%$, for all label scenarios and achieves a negative recovery score $R_d = -29\%, -27\%, -23\%$. On **FMoW** and **Yearbook**, the differences between the accuracy of Naïve, S4L and PL are negligible as the largest improvement over Naïve is $+2.3\%$ on Yearbook under the large label delay scenario $d = 100$.

**TTA.** In Figure 3, we find that TTA consistently under-performs every method, *including* the delayed Naïve, under every delay scenario on the CLOC, CGLM and FMoW datasets Nevertheless, on Yearbook TTA successfully outperforms IWMS, S4L, PL and Naïve by up to $+1.7\%$ in the moderate label delay scenario $d = 50$. Over the four dataset, the exact extent of the recovery of the accuracy gap $R_d$ for $d = 10, 50, 100$, respectively, is as follows: on **CLOC** $R_d = -87\%, -44\%, -36\%$, on **CGLM** $R_d = -77\%, -67\% - 62\%$, on **FMoW** $R_d = \underline{-480\%}, \underline{-227\%}, -159\%$ and on **Yearbook** $R_d = 4\%, 11\%, 11\%$. The disproportionately severe negative result on FMoW is due to the otherwise small accuracy gap $G_d = -2.5\%, -5.2\%, -7.4\%$. More importantly, we hypothesize that TTA fails to outperform Naïve because the common assumptions, upon which TTA methods were designed, are broken.

## 7 Analysis of Importance Weighted Memory Sampling

We first perform an ablation study of our IWMS to show the effectiveness of the importance sampling. Then, we show our performances under different computational budgets and buffer sizes. We provide further in-depth analysis of the information retention abilities of the considered methods in the Supplementary Material A.13 and the effect of memory size in A.14.

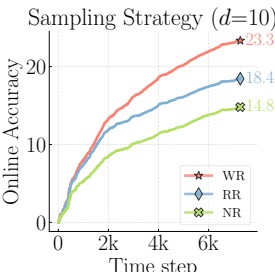
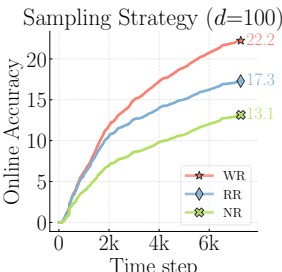

Figure 4: **Effect of sampling strategies** We report the Online Accuracy under the least ( $d = 10$ ) and the most challenging ($d = 100$) label delay scenarios on CGLM [5].

**Analysis on Memory Sampling Strategies.** Note that while our method, IWMS is a prioritised sampling approach, it has some similarities to Naïve, except for the sampling strategy. While the Naïve method uses the most recent labeled data and a randomly sampled mini-batch from the memory buffer for each parameter update, our method provides a third option for constructing the training mini-batch, which picks the labeled memory sample that is most similar to the unlabeled data. When comparing sampling strategies, we refer to the newest batch of data as (N), the random batch of data as (R) and the importance weighted memory samples as (W).

In Figure 4, we first show that in both delay scenarios ($d = 10$ and $d = 100$) replacing the newest batch (N) with (W) results in almost doubling the performance: $+8.5\%$ and $+9.1\%$ improvement over Naïve, respectively. Interestingly enough, when we replace the (N) with uniformly sampled random buffer data (R) we report a significant increase in performance. We attribute this phenomenon to the detrimental effects of label delay: even though Naïve uses the most recent supervised samples for training, the increasing discrepancy caused by the delay $d = 10$ and $d = 100$ forces the model to over-fit on the outdated distribution.

# 8 Conclusion and Future Work

We motivate modeling real-world scenarios by introducing the label delay problem. We show how severely and unpredictably it hinders the performance of approaches which *naïvely* ignore the delay. To address the newfound challenges, we adopt the three most promising paradigms (Pseudo-Labeling, S4L and TTA) and propose our own technique (IWMS). We provide extensive empirical evidence over four large-scale datasets posing various levels of distribution shifts, under multiple label delay scenarios and, most importantly, under normalised computational budget. IWMS simply stores and and reuses the embeddings of every observed sample during *memory rehearsal* where the most relevant labeled samples to the new unlabeled data are rehearsed. Due to its simplicity, the robustness against changes in the data distribution can be implemented very efficiently.

# 9 Acknowledgement

Botos Csaba was partially funded by Intel and partially by Meta AI Research. This work is supported by a UKRI grant Turing AI Fellowship (EP/W002981/1) and EPSRC/MURI grant: EP/N019474/1. Adel Bibi acknowledges the funding from the KAUST Office of Sponsored Research (OSR-CRG2021-4648) and the support from Google Cloud through the Google Gemma 2 Academic Program GCP Credit Award. The authors thank Razvan Pascanu and João Henriques for their insightful feedback. We also thank the Royal Academy of Engineering.

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

# A Supplementary Material

## A.1 Dataset Statistics

We conduct our experiments on four large-scale online continual learning datasets, Continual Localization (CLOC) [4], Continual Google Landmarks (CGLM) [5], Functional Map of the World (FMoW) [6], and Yearbook [7]. The last two are adapted from the Wild-Time challenge [41]. More statistics of the benchmarks are in Supplementary.

The first, Continual Localization (CLOC) [4] which contains 39M images from 712 geolocation ranging from 2007 to 2014. The second is Continual Google Landmarks (CGLM) [5] which contains 430K images over 10788 classes. Followed by that, we report our experiments on Functional Map of the World (FMoW) [6] adapted from the Wild-Time challenge [41]. The dataset contains 14,696 satellite images, from 2002 to 2017, with the task of predicting the land type. Last, we show our results on the Yearbook dataset [7] containing 33,431 frontal-facing photos from American high-school yearbooks. The photos were taken in the time-period between 1930-2013 and represent changes in fashion, gender and ethnicity over the years. The task is a binary classification problem: predicting the gender of the student based on the photo.

## A.2 Implementation Details of S4L

For integrating S4L methods, we adopt the most effective approach through iterative optimization of both supervised and unsupervised losses. This process involves optimising the standard Cross Entropy loss on labeled data (similar to Naïve) and minimising contrastive loss on unlabeled data, utilising a balanced approach until exhausting the computational budget. We conducted an exhaustive search over the possible multi-objective optimisation variants (such as iterative and joint optimisation) and determined the best result is achieved when the contrastive loss is minimised separately for the first half of the parameter update steps, followed by minimising the supervised loss for the second half of the update steps. We report the best results across the three main families of contrastive losses, i.e.,, Deep Metric Learning Family (MoCo [38], SimCLR [44],and NNCLR [45]), Self-Distillation (BYOL [46] and SimSIAM [47], and DINO [40]), and Canonical Correlation Analysis (VICReg [48], BarlowTwins [49], SWAV [50], and W-MSE [51]).

For fair comparison, we normalise the computational complexity [8, 9] of the compared methods. We find that while SSL methods may take multiple forward passes, potentially with varying input sizes, the backward pass is consistently done only once among the variants, therefore, we choose the number of backward passes to measure the computational complexity of the resulting methods. According to this computational complexity constraint, Naïve augmented with SSL at each time step takes two backward passes, one for computing the gradients of the Cross Entropy over the labeled samples and one for the Contrastive Loss over the unlabeled samples, thus $\mathcal{C}_{\text{S4L}} = 2$.

## A.3 Monotonous Online Accuracy Degradation

We argue the persistent drop in the Online Accuracy is due to the non-stationary property of the data distribution that creates a distribution shift. Our hypothesis is supported by the experimental results, illustrated in Figure 5: the Online Acc gradually decreases as the function of label delay $d$, at any given time step $t$. Furthermore, in Figure 6, we summarize the *final* Online Accuracy scores, i.e., the Online Accuracy value at the final time step of each run

Our claims are reinforced by the findings of Yao et al. [41]. A complementary argument is presented by Hammoud et al. [43], stating that the underlying datasets have high temporal correlations across the labels, i.e., images of the same categories arrive in bursts, allowing an online learning model to easily over-fit the label distribution even without using the input images.

## A.4 Qualitative Analysis of Label Delay

**A case study of the distribution shift in the Yearbook experiments.** While *Online Accuracy* is a well established performance metric for Online Continual Learning [4, 8, 5, 43], it can conceal some of the most important characteristics of the underlying dataset. To highlight a direct connection between the distribution shift and its immediate impact on the model performance, we illustrate the

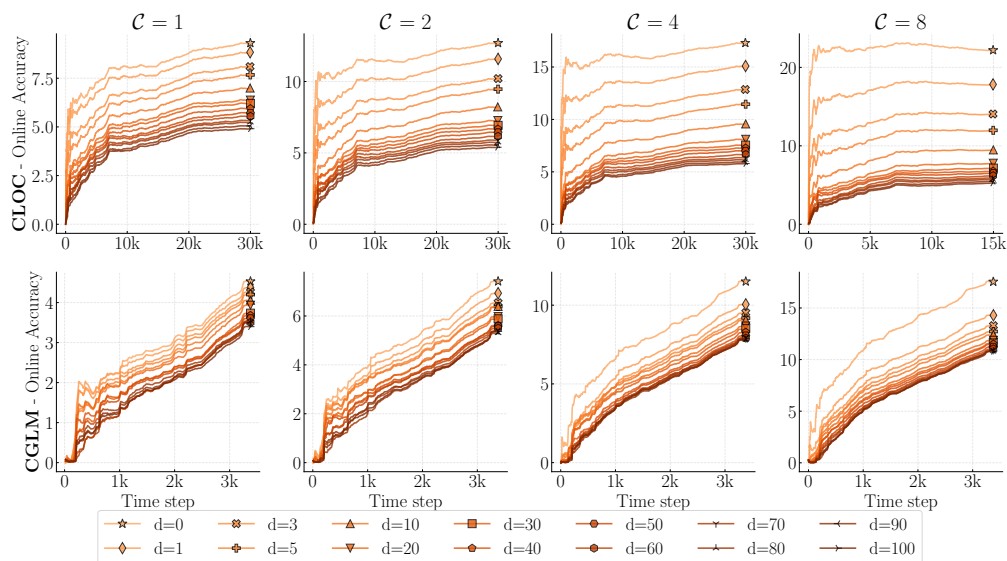

Figure 5: **Monotonous degradation of Online Accuracy** with regards to label delay $d$, over multiple datasets, CLOC [4] and CGLM [8], under various computational budgets, $\mathcal{C} = 1, 2, 4, 8$. The accuracy gradually drops at *every* time step $t$ as the function of the label delay $d$. However the extent of the degradation is non-linear: The initial smallest increases in label delay have severe impact on the performance. In contrast, the rate of degradation slows down even for an order of magnitude larger increments when the labels are already delayed. See Figure 6 for the summary of the final values.

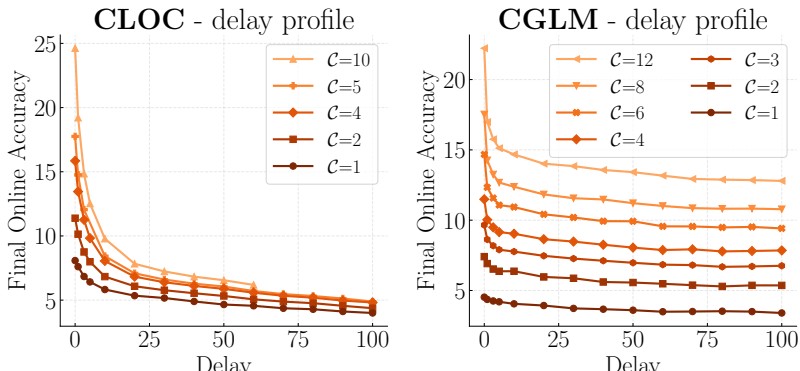

Figure 6: **Delay Profile.** Each trajectory shows the *Final Online Accuracy*, i.e., the Online Accuracy evaluated at the last time step of each run, at a fixed computational budget $\mathcal{C}$. On both datasets the most severe accuracy degradation occurs in the first quarter ($d = 0 \rightarrow 25$). In contrast, on CGLM [8], the degradation is not significant in lower compute regimes $\mathcal{C} \leq 4$.

Top-1 Accuracy of the *current* batch at each time step in Figure 7. The experimental settings are identical to the main experiments on Naïve, detailed Section 5.2.

In this experiment, we describe several observations: first, the models perform at per-chance level accuracy until the first batch of labeled data arrives. Notice that the per-chance level is not 50% because the dataset is biased (contains more male than female portraits). However as the ratio improves over time, the random classifier's accuracy gets closer to 50%.

**Before the distribution shift.** In the smallest delay scenario (yellow curve), the delay is identical to a lag of three years between making the predictions and receiving the labels. Under such delay, the model quickly reaches close-to-optimal accuracy under just a few time steps and performs identically

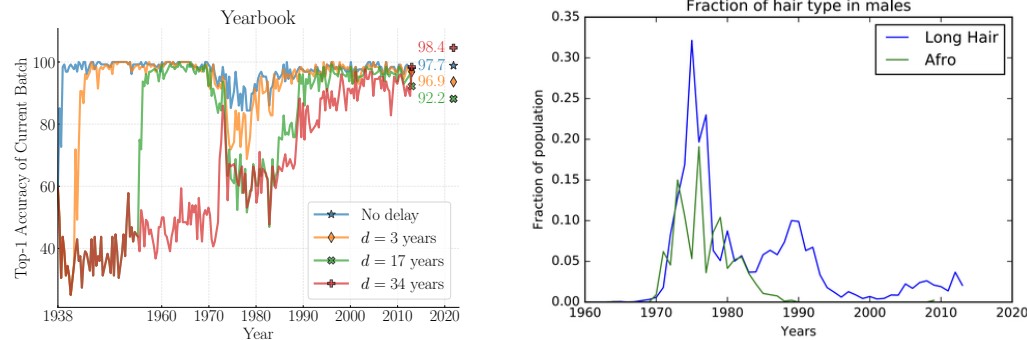

Figure 7: **(Left)** Top-1 Accuracy of Naïve on the *current* batch (of time step $t$) of Yearbook. **(Right)** Report from Ginosar et al. [7] on "the fraction of male students with an afro or long hair." The drop in Top-1 Accuracy over time strongly correlates with the change in appearance of one of the two classes in the Yearbook [7] dataset. The larger the delay, the longer it takes to recover the close-to-perfect accuracy.

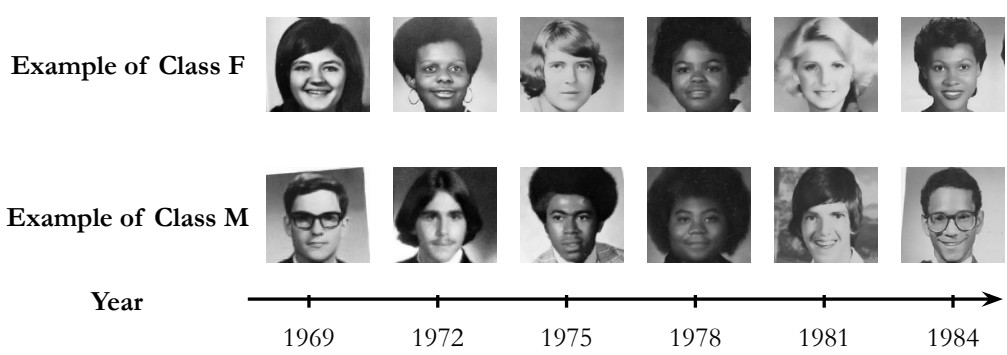

Figure 8: Examples from the Yearbook dataset [7] during the time where the visual appearance of men (bottom row) changes drastically resulting in an accuracy drop of an online classifier, regardless of the label delay.

to the non-delayed counterpart (blue curve). In the moderate delay scenario (green curve), the model stays "idle" for a longer time (equivalent of 17 years) because of the delay of the first labeled batch. Nevertheless, the delayed model reaches similarly good performance after a time steps. Interestingly, the severely delayed model (red curve), exhibits a steep increase in performance, at $t = 1972$, exactly 34 years after observing the first sample ($t = 1938$).

**During the distribution shift.** The steep increase in the most severely delayed scenario (red curve) coincidentally overlaps with a major distribution shift in the appearance of one of the two classes. This shift simultaneously impacts the performance of all four models, however the rate at which their performance recovers differs, due to the label delay. While in general it is an immensely difficult problem to detect and trace the changes of the data distribution, due to hidden latent variables (such as socio economic factors, genetic diversity of the population, cultural and political trends), Ginosar et al. [7] identified and tracked many of such variables. One of these factors, namely the "fraction of male students with an afro or long hair", is highly correlated (in the temporal dimension) with the accuracy drop in our experiments, as illustrated in the right-hand side of Figure 7.

**The reason behind the accuracy drop.** In the qualitative experiments of the section titled *"What time specific patterns is the classifier using for dating?"*, Ginosar et al. [7] reports that convolutional neural networks, such as VGG [53], learn to extract features from the hairstyles of the subjects. Although the task is slightly different, classification of the year of the photograph, we hypothesise

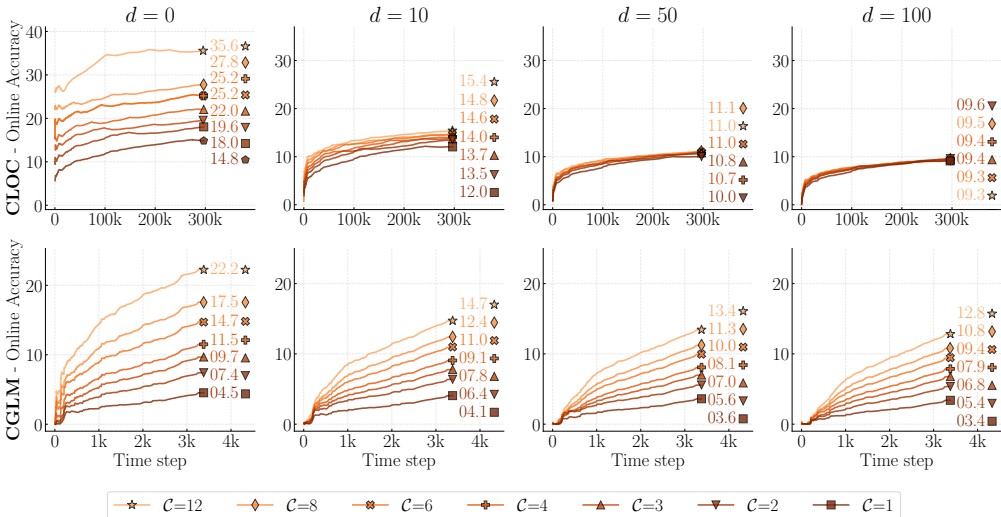

Figure 9: **Diminishing returns** of increasing the computational budget $\mathcal{C}$ over four label delay regimes $d = 0, 10, 50, 100$, on two datasets. While in many real-world scenarios simply increasing the budget $\mathcal{C}$ to improve the overall performance, when the labels are delayed the improvements *may* become marginal. Interestingly, this phenomena is emphasized on the CLOC [4] dataset, as the trajectories collapse to a single curve as the delay increases $d = 0 \to 100$. In contrast, on CGLM [8] the relative improvements, i.e., the vertical distances between the lines, may shrink going from $d = 0 \to 10$, but stay consistent for $d = 10 \to 100$. The final scores are summarized by Figure 10.

that one of the most discriminative features learned by the model are related to the hairstyles, as it is the most influential variable in terms of the accuracy of four independently trained models.

**After the distribution shift.** The recovery of the accuracy can be characterised by two factors: 1) the severity of the level degradation and 2) the duration of the recovery. Both factors show strong dependency on the underlying label delay factor: the larger the delay the larger the degradation and the longer the recovery length. Notice how closely the slightly delayed, yellow curve ($d = 3$ years) follows the non-delayed, blue curve in terms of duration, while the extent of the accuracy drop is larger for the delayed counterpart. On the other hand, the moderately and severely delayed models (green and red curves, respectively) apparently reach a lower-bound in performance degradation, where larger delay does not further reduce the accuracy. Nevertheless, the recovery of the severely delayed model is slower and occurs later than the moderately delayed model.

## A.5 The impact of label delay on the scaling property of the computational budget

The exploration of the impact of label delay on computational efficiency and accuracy across different settings reveals important insights into the performance and scalability of *Naïve*, an Experience Replay model [26], which simply waits for every sample to receive its corresponding label before using it as a training data. In this section, through extensive **quantitative comparison** under different label delay $d$ and computational budget $\mathcal{C}$ regimes, we offer a comprehensive overview of how these key factors interact to influence model performance on two large-scale datasets: CLOC [4] and CGLM [8].

**Diminishing Returns.** Figure 9 highlights the phenomenon of diminishing returns on investment in the computational budget $\mathcal{C}$ across four different label delay regimes ($d = 0, 10, 50, 100$). Notably, while augmenting $\mathcal{C}$ typically yields performance improvements, these gains become increasingly marginal in the presence of delayed labels. The impact of label delay is markedly pronounced in the CLOC dataset, where the performance trajectories converge into a singular trend as the delay escalates from $d = 0$ to $d = 100$. Conversely, the CGLM dataset exhibits a contraction in the relative improvements (vertical distances between performance trajectories) as delay transitions from $d = 0$ to $d = 10$, yet these differences remain relatively stable for delays extending from $d = 10$ to $d = 100$.

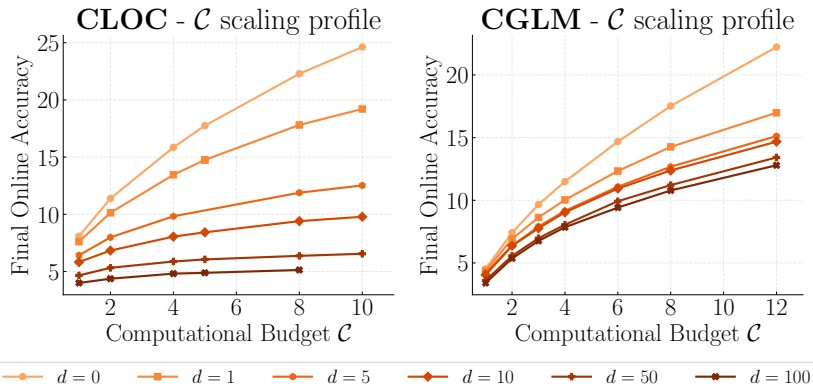

Figure 10: **Compute Scaling Profile.** Each trajectory shows the *Final Online Accuracy*, i.e., the Online Accuracy evaluated at the last time step of each run, at a fixed computational budget $\mathcal{C}$. We show sub-linear improvement w.r.t. subsequent increases in $\mathcal{C}$, even in the non-delayed ($d = 0$) scenario. Moreover, the influence of label delay on the scaling property varies between the two datasets: while on CLOC [4] large delays ($d = 100$) prevent the model from benefiting from more parameter updates, on CGLM [8] label delay (for $d > 1$) only seems to offset the Final Online Accuracy, but does not impact rate of improvement.

**Compute Scaling Profile.** In Figure 10, the concept of a Compute Scaling Profile is introduced, displaying the *Final Online Accuracy* – the accuracy measured at the last time step of each run – for various levels of computational budget $\mathcal{C}$. This figure elucidates the sub-linear scaling of performance improvements with respect to incremental increases in $\mathcal{C}$, a trend observable even without label delays ($d = 0$). The effects of label delay diverge between the datasets; CLOC sees a significant impediment to performance gains from additional parameter updates at high delays ($d = 100$), while in CGLM, the delay primarily shifts the Final Online Accuracy without diminishing the rate of improvement.

**Gradual Monotonous Degradation.** Figure 5 presents a nuanced view of how Online Accuracy monotonically degrades with increasing label delay ($d$) across different computational budgets ($\mathcal{C} = 1, 2, 4, 8$). This degradation is not linear; initial increments in label delay incur a steep decline in performance, whereas the rate of decline moderates for larger increments of delay, showcasing a nonlinear impact on model accuracy over time.

**Delay Profile.** Finally, Figure 6 encapsulates the Delay Profile, depicting the Final Online Accuracy at various computational budgets ($\mathcal{C}$). Both datasets exhibit the most substantial accuracy reductions in the initial quarter of delay increments ($d = 0 \rightarrow 25$). Interestingly, the CGLM dataset demonstrates a negligible degradation in lower computational regimes ($\mathcal{C} \leq 4$), indicating a potential resilience or adaptive capability under specific conditions.

While increased computational budget generally improves the performance, the presence of label delays introduces a complex dynamic that can significantly hinder these benefits. The distinct behaviors observed across the CLOC and CGLM datasets further suggest that the dataset characteristics play a pivotal role in the decision making whether investment in additional compute is warranted or not. We suggest that such decision should be made on a case by case basis, rather than extrapolating from publicly available benchmarks.

## A.6 Breakdown of SSL methods

In Figure 11 we show the performance of the best performing SSL based methods after hyper-parameter tuning. We observe that the performance of the SSL methods is highly dependent on the dataset and the delay setting. However, we apart from MoCo v3 [39], the methods perform similarly to Naïve on CLOC. On the other hand on CGLM they have insignificant differences in performance, but consistently underperform Naïve.

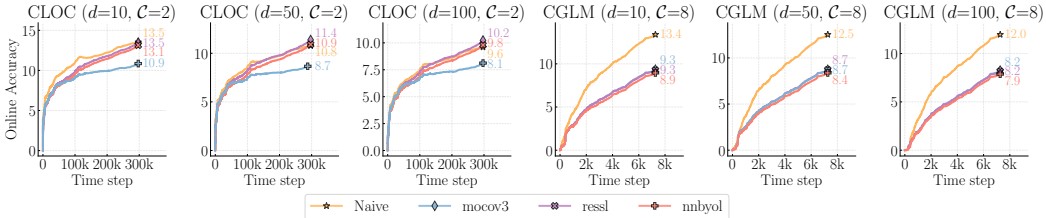

Figure 11: Comparison of the best performing SSL based methods after hyper-parameter tuning

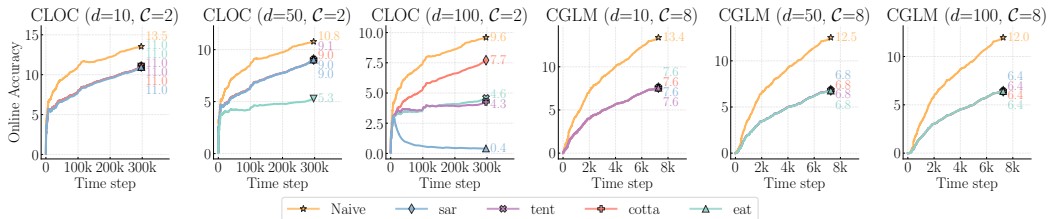

Figure 12: Comparison of the best performing TTA based methods after hyper-parameter tuning

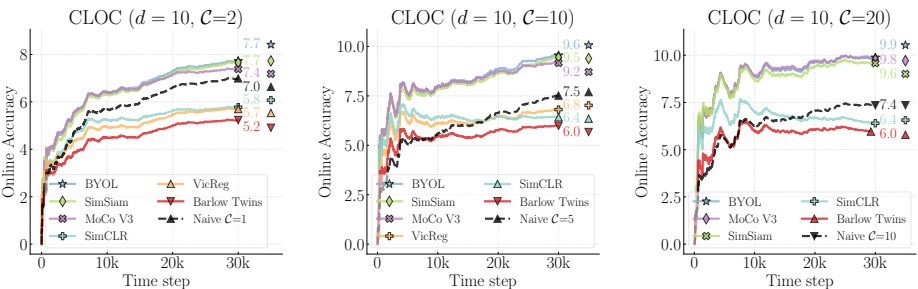

Figure 13: Detailed breakdown of various Self-Supervised Learning methods from each family. Results are shown across varying number of parameter updates $\mathcal{C} = 2, 10, 20$ under the $d = 10$ scenario.

## A.7 Breakdown of TTA methods

In Figure 12 we show the performance of the best performing TTA based methods after hyper-parameter tuning. We observe that the performance of the TTA methods are consistently worse than Naïve on both CLOC and CGLM, under all delay settings. We observe that in the most severe delay scenario ($d = 100$) the performance of EAT [52] and SAR [35] is comparable to Naïve on CLOC, while CoTTA [37] avoids the catastrophic performance drop.

## A.8 Comparison of S4L to Naïve when using the same amount of supervised data

While in our main experiments S4L fails to outperform Naïve (in Section 6), we show that it is mostly due to the computational constraint of our experiments. In order to test our hypothesis, we run a series of experiments on the S4L variants, illustrated in Figure 13. In this experiment, instead of limiting the Computational Budget $\mathcal{C}$, we directly restrict the number of parameter updates to test if optimising the joint objective of Naïve and the given Self-Supervised Learning method improves the performance of the model at all. Our results indicate positive improvement over Naïve for MoCo-V3 [39], SimSiam [47] and BYOL [46] consistently across multiple settings with increasing number of parameter updates.

First, on the **left** hand side of Figure 13, both the Naïve and the S4L variants take only a single parameter update per time step (thus $\mathcal{C} = 2$ for all, except Naïve, where $\mathcal{C} = 1$). On the first 10%

of the CLOC dataset [4], this results in a modest, nevertheless clear improvement over Naïve, up to $+0.7\%$. Followed by that, in the **middle**, every model takes five parameter updates per time step. Notice that Naïve has a stricter computational budget, $\mathcal{C} = 5$, to match the rest of the experiments. Consistently with our findings in Section A.5, Naïve only benefited marginally from the increase in compute, due to diminishing returns, $7.0\% \rightarrow 7.5\%$. On the contrary, the previously highlighted S4L variants show a larger improvement over the increase in number of updates, e.g., $7.7\% \rightarrow 9.6\%$. Consequently, this increases the gap between the Naïve and the S4L methods. Finally, on the **right** hand side of the figure, we show when the models are updated ten times in each time step, the improvement plateaues for both the Naïve and the S4L variants.

**Conclusion** of this set of experiments is two-fold: when granted equal amount of parameter updates, S4L methods outperform Naïve across different settings. However, computing the parameter gradients w.r.t. the joint objective of S4L costs approximately twice the amount that of the Naïve: $\mathcal{C}_{\text{S4L}} \simeq 2 \times \mathcal{C}_{\text{Naïve}}$. Due to the well-known property of Self-Supervised Learning methods, *sample inefficiency*, our main experiments show that "spending" the compute on more frequent Naïve updates is more beneficial than optimising the joint S4L objective, even when the training data is heavily delayed.

### A.9 Examples of the Importance Weighted Memory Sampling on CLOC

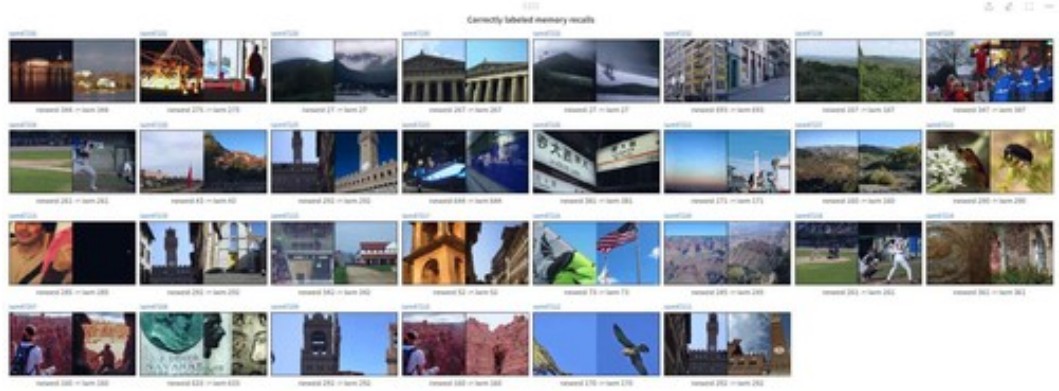

Figure 14: **Correctly labeled memory recalls.** In the subfigure's caption "Newest" refers to the newest unsupervised image observed by the model and "iwm" refers to the sample drawn from the memory by our proposed sampling method. The numbers refer to the corresponding true label IDs.

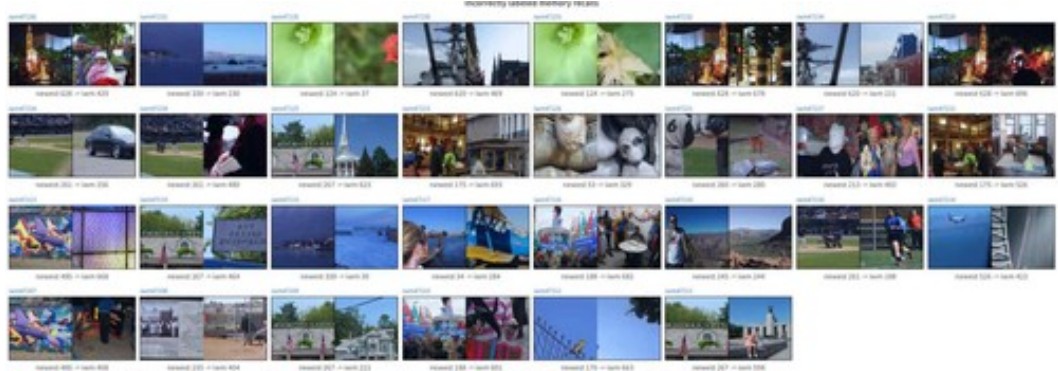

Figure 15: **Incorrectly labeled memory recalls.** In the subfigure's caption "Newest" refers to the newest unsupervised image observed by the model and "iwm" refers to the sample drawn from the memory by our proposed sampling method. The numbers refer to the corresponding true label IDs.

On CLOC, we report similar scores to Naïve due to high noise in the data. To provide evidence for our claims we visualize the supervised data sampled from the memory buffer by our Importance

Weighted Memory Sampling method. In Figure 14, we show that our method is capable of guessing the correct location of the unsupervised sample (the left hand side of the image pairs) and recalling a relevant sample from memory. In contrast, the incorrect memory recalls hurt the performance even though the content of the samples might match. We illustrate such cases in Figure 15, where it is obvious that in some cases the underlying image content has no information related to the location where the picture was taken at. In such scenarios, the only way a classifier can correctly predict the labels is by exploiting label correlations, e.g., classifying all close-up images of flowers to belong to the same geo-location, even though the flowers are not unique to the location itself. Or consider the pictures taken at social gatherings (second row, second column from the right), where a delayed classifier without being exposed to that specific series of images has no reason to correctly predict the location ID. Our claims are reinforced by the findings of [43].

## A.10   Visual Explanation of our Experimental Framework

We provide visual guides for explaining our experimental framework. In Figure 16, we emphasize the main difference between our setting and the general setting of partially labeled data-streams: while prior art does not differentiate between old and new unsupervised data, our work focuses specifically on the scenario when *all* unsupervised data is newer then the supervised data. In Figure 17, we show the two types of data that our models work with: outdated supervised data, and newer, unsupervised data. The task is to find a way to utilize the newer unsupervised data to augment the Naïve approach, that simply just waits for the labels to become available to update its parameters. The most challenging component in our experiments is the computational budget factor that allows only a certain amount of forward and backward passes through the backbone.

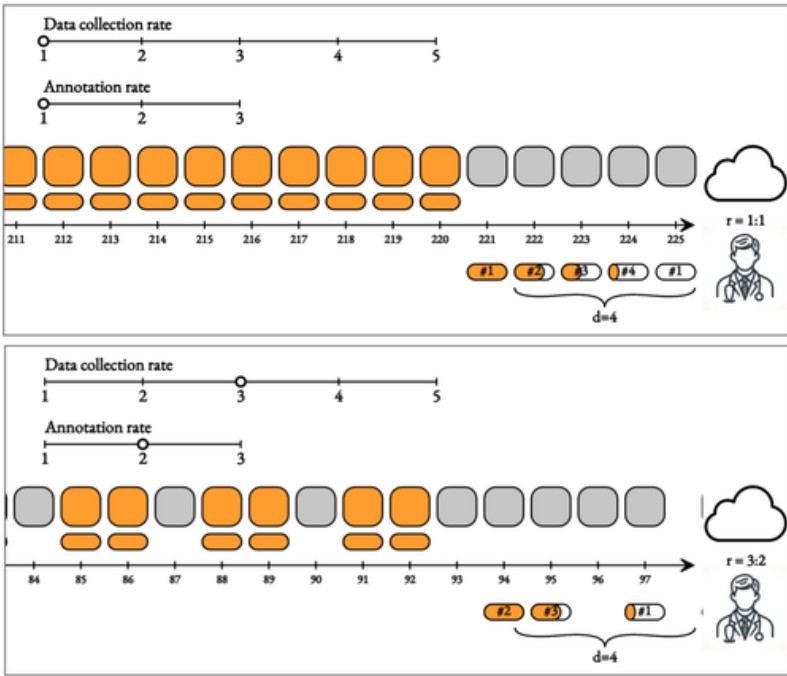

Figure 16: **Our experimental setup (top)**: After a fixed amount of time steps all labels become available. This allows us to focus on utilizing future unsupervised samples effectively. **Partial labeling setup (bottom)**: in the generic setting, when the data collection rate is higher than the annotation rate, some samples might never receive labels.

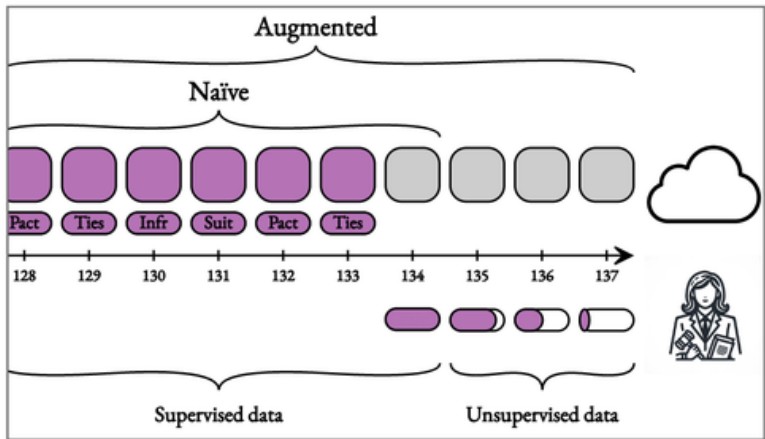

Figure 17: **Experimental setup**: in our experiments we show how increased label delay affects the Naïve approach that simply just waits for the labels to arrive. To counter the performance degradation we evaluate three paradigms (Self-Supervised Learning, Test-Time Adaptation, Importance Weighted Memory Sampling) that can augment the Naïve method by utilizing the newer, unsupervised data.

## A.11 Two-stage vs single-shot sample selection

In Section 4, we outlined our proposed two-stage sample selection method, IWMS. In this experiment we show empirical evidence and analysis on why predicting the class-labels first then doing similarity matching leads to better results than simply using a similarity score over all the memory samples. In Figure 18, we illustrate the evolution of the similarity scores of the two matching policies. On the left, the matching is done purely based on the similarity scores, whereas on the right only those samples were compared against those memory samples whose labels match the predicted labels. In the middle plot, we show that by implementing the two-stage selection, we increase the effectivity of the similarity matching by a large margin, $+7.8\%$.

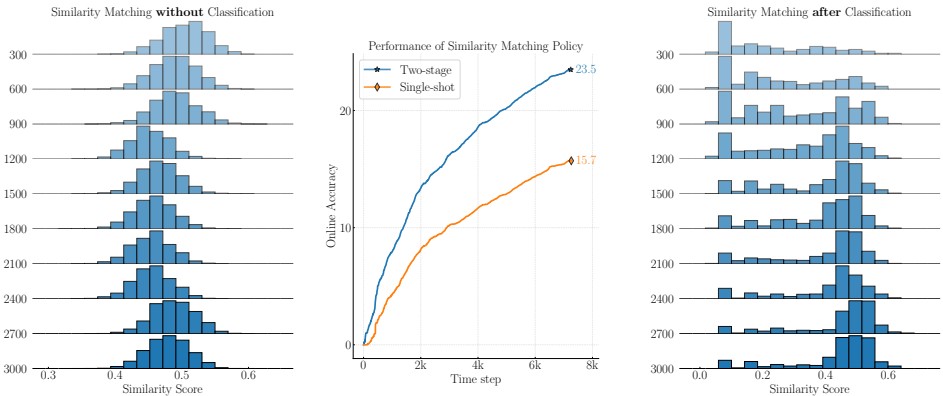

Figure 18: The evolution of Similarity Scores between the unsupervised and memory samples over time. On each histogram, we plot the distribution of the cosine similarity scores between the feature representations of the yet to be labeled samples and the samples in the memory that already received their labels. On the top row we show the initial distributions and going from top down, the evolution of the two distribution is illustrated over the time steps.

### A.12 Extended Literature Review on Online Learning

**Online Learning vs Online Continual Learning**: Online Learning and Online Continual Learning, while both involve learning from data arriving sequentially, differ fundamentally in scope. Online Learning typically deals with single-task streams, often assumed to be from an i.i.d. distribution, as outlined in section 2.3 of [54] and the introduction of [55]. In contrast, Online Continual Learning (OCL) is more concerned with non-stationary streams that undergo frequent changes in distribution, where mitigating forgetting is one of several challenges [55, 56, 57].

**Non-i.i.d. distribution of unsupervised data:** While our work focuses on evolving distributions, work such as Weinberger et al. [13] and Flaspohler et al. [58] only considers label delay while the distribution a time-invariant, consequently completely omitting the problem of distribution shift. Majority of the prior online learning work [41, 14, 11, 20, 19, 15, 59, 13, 17, 18, 58] ignores the difference between past and future unsupervised data. In our proposal, all unsupervised data is newer than the last supervised data. We illustrate the difference between the two different types of unsupervised data in Figure 16.

**Considering catastrophic forgetting:** Continual Learning, both online and offline, is concerned about performing well on previously observed data, often referred to as backward transfer of the learned representations [41, 14, 11, 20, 19, 15, 59, 13, 17, 18, 58]. This is different from Online learning where the problem of forgetting is not considered. Even in more recent Online Continual Learning work, backward transfer have been given slightly lower priority [43, 9, 4] where the authors have reported them only in the appendix.

Furthermore, most of the prior art does not differentiate between the past and future unlabeled data. In our proposal, all unlabeled data is newer than the last labeled data due to delayed annotation, as illustrated in Figure 16. RealtimeOCL also considers *delay*, however, their delay arises from model complexity; in their *fast-stream* scenario, the stream releases input-label pairs quicker than models can update, causing models to be trained on an older batch of samples. In essence, labels are still instantly available in RealtimeOCL, while our work examines delay attributed to the non-instantaneous arrival of labels. RapidOCL [43] highlighted the exploitation of label-correlation in online continual learning, with a focus on measuring online accuracy through future samples. In contrast, our framework allows the models to leverage the more recent, unlabeled data for adaptation. While a growing line of work adapts S4L to continual learning to make use of unlabeled data in continual learning settings, such as CaSSLe [60] in task-agnostic settings and SCALE [61] in task-free settings, most previous work did not perform a comprehensive examination of PL and S4L under a strict computational budget.

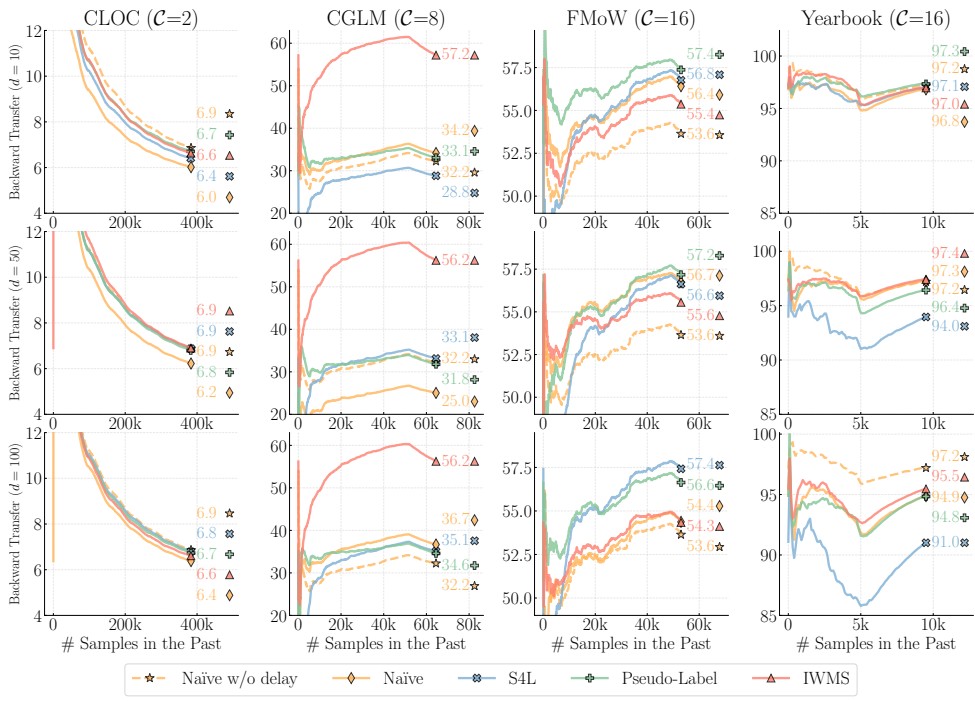

Figure 19: **Backward transfer.** Measuring forgetting on the withheld validation set.

## A.13    Analysis on forgetting over past samples

In Figure 19, we report the backward transferability of the learned representation. This is done on a held-out, ordered validation set where the timestamp is used for ordering. On **CLOC**, all methods perform similarly due to poor data quality as reported in the Supplementary Material A.9. On **CGLM**, our method not only surpasses the performance of others, but achieves $\sim 2\times$ the accuracy of the S4L, PL, Naïve and non-delayed Naïve baseline on CGLM. This means that the representation learned by our sampling technique is far more robust and generalises better not only to future but past examples as well. On **FMoW**, the best result is achieved by the Semi-Supervised methods, nevertheless our method outperforms the non-delayed Naïve in all scenarios. Finally, on **Yearbook** we see that under low label delay ($d = 10$) all results are clustered around 97%, however IWMS and Naïve performs best under larger delays ($d = 50, 10$).

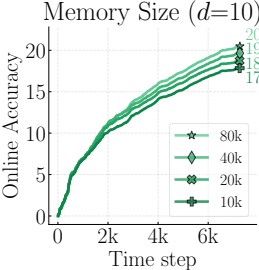
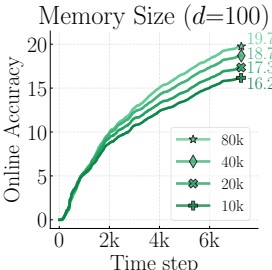

Figure 20: **Effect of memory sizes (right).** We report the Online Accuracy under the least (top: $d = 10$) and the most challenging (bottom: $d = 100$) label delay scenarios on CGLM [5].

### A.14 Analysis on the Memory Size.

We study the influence of buffer size on our proposed IWMS. In particular, we show the performance of our algorithm under the buffer size from 10K to 80K in Figure 20. Even though IWMS relies on the images sampled from the buffer to represent the new coming distribution, its performances remain robust under different buffer sizes: the largest performance gap between memory sizes of 10K and 80K is a marginal 2.5%.

### A.15 TTA Normalized Computational Budget Considerations

We report several TTA methods, such as CoTTA [37] and SAR [35], which abuse the absence of formal computational constraints in traditional Test-Time Adaptation settings by computing the entropy of the predictions of the input data up to $32\times$ different augmentations. Methods, such as EATA [52] further complicate the complexity normalisation problem by using multiple smaller-sized crops of the input image. To simplify our comparisons, we ignore the cost of model inference, thus $\mathcal{C}_{\text{TTA}} = 1$.

### A.16 Training Time

Our computation budget $\mathcal{C}$ is based on the number of forward-backward passes, which is generally a good proxy for time and it has been widely used in CL literature [25, 8, 9, 43]. The actual wall-clock training time can be influenced by various factors, e.g., code optimization, hardware, data I/O speed, and implementation.

Here we add the training time for our method, ER, contrastive learning-based method, pseudo-labeling based method, and TTA methods with the same number of forward-backward passes in Table 1. Most of the experiments are using a single A100 GPU with 12 CPU. The training time is measured in hours. This table shows that the training time could be entirely different due to various other factors.

Table 1: Training times (in hours) for various methods across different datasets. [1] The CPU allocation was 6.

|                 | CLOC     | CGLM | FMoW | Yearbook |
|-----------------|----------|------|------|----------|
| Naive           | 52       | 3.6  | 2.3  | 0.20     |
| ReSSL           | 67       | 3.0  | 4.0  | 0.25     |
| CoTTA           | 39       | 5.0  | 2.5  | 0.20     |
| Pseudo Labeling | 111[1]   | 4.6  | 2.5  | 0.15     |
| IWMs            | 61       | 3.6  | 3.5  | 0.20     |

### A.17 Alternative Conventional Metrics

In the Continual Learning literature a pletora of metrics have been proposed, of which the most popular are Average Accuracy [62], Task based Forward and Backward Transfer [62, 25], Model

Plasticity [63] which entails *Learning Accuracy* and *Learning Forgetting*. While most of these metrics aim to capture and aggregate semantically similar aspects of the observed learning dynamics, they have a slight variation in the definition.

In Mai et *al.* [62], under the "Evaluation Metrics", Average Accuracy is defined as follows:

$$\text{AverageAccuracy}(A_i) = \frac{1}{i} \sum_{j=1}^{i} a_{i,j} \tag{1}$$

which is closely related to Online Accuracy, but we appreciate the fact that in such a scenario, performance on past iterations is re-evaluated in every step. In our setting, evaluating and reporting this metric would be infeasible, as the number of steps on the CLOC, CGLM, and FMoW datasets is 3-to-4 orders of magnitude larger than the examples provided in the survey (the maximum number of steps in [62] is 20; for comparison, in our experiments the maximum number of "tasks" is 296,119).

With a slight variation, Guo et al. [64] details the evaluation metric in section 6.1 as follows:

> We first learn from the data stream of all tasks for each dataset, and then test the final model using the test data of all tasks. We report the average accuracy of all tasks from 15 random runs for each dataset.

We argue that this metric is taking an excessively strong measure to remove noise from the metric. In our experiments, we experienced that rerunning the same training with different seed results in negligible (less than 0.01%) differences in the results. Running the experiments 15 times to evaluate the metric of [64] is infeasible for us.

This similarly holds for [65] as well, since they report the Average Accuracy and Average Forgetting across 15 runs.

Furthermore, although the main manuscript of [66] does not provide the detail about re-running the Average Accuracy, the corresponding code is set by default to 15 re-runs with different seeds. (follow this URL for reference: https://github.com/YananGu/DVC/blob/6f12984d10a4a1c4609f221b939f93d94fc8258e/general_main.py#L29)

In [67], the number of random initializations is dropped to 10, otherwise they report the Average Accuracy as well.

Koh et al. [68] introduces their own metric: Knowledge Loss/Gain Ratio, claiming that the metrics used by [64, 65, 66] are relying on the notion of task boundaries therefore they define a new objective that is "appropriate for periodic data distribution". In our paper we cannot make such assumptions about periodicity.

The accuracy metric proposed by Wang et al. [63], Learning Accuracy (LA) using **Model Plasticity** is formally defined for the $j$-th task as:

$$l_j = a_j^j \tag{2}$$

where $a_j^j$ is the accuracy evaluated on the test set of task $j$ after training the network from task 1 to task $j$. We would like to argue that this metric is similar to the *Online Accuracy* metric apart from the notion that here the test samples are drawn from a different distribution, whereas the Online Accuracy is evaluated on the $j$-th batch of data before it is used for training. If we assume that both the test and the training batch is drawn from the same distribution at time-step $j$, the two metrics are arguably the same. (Please note that the training is only done on the batch after the evaluation in the case of Online Accuracy.)

To ensure that all relevant metrics can be computed for future reference, we run the Naïve and IWMS experiments on the two datasets where IWMS was performing the best and the worst to provide a full comparison against the baseline.

We simplified the table representation by splitting the validation data into 100 equal sized ranges along the time axis, such that the ranges would correspond to the training data range:

Table 2: Accuracy matrix for Naïve method on Yearbook dataset.

| Accuracy | $te_{0\to12}$ | $te_{12\to25}$ | $te_{25\to37}$ | $te_{37\to50}$ | $te_{50\to62}$ | $te_{62\to75}$ | $te_{75\to87}$ | $te_{87\to100}$ |
|---|---|---|---|---|---|---|---|---|
| $tr_{0\to12}$ | 0.99 | 0.98 | 0.94 | 0.76 | 0.56 | 0.70 | 0.89 | 0.88 |
| $tr_{12\to25}$ | 0.98 | 0.99 | 0.97 | 0.79 | 0.58 | 0.72 | 0.88 | 0.86 |
| $tr_{25\to37}$ | 0.99 | 0.99 | 0.96 | 0.75 | 0.55 | 0.59 | 0.77 | 0.86 |
| $tr_{37\to50}$ | 0.99 | 1.00 | 0.99 | 0.83 | 0.65 | 0.74 | 0.87 | 0.90 |
| $tr_{50\to62}$ | 0.94 | 0.96 | 0.96 | 0.88 | 0.88 | 0.94 | 0.96 | 0.94 |
| $tr_{62\to75}$ | 0.97 | 0.99 | 0.99 | 0.96 | 0.93 | 0.93 | 0.97 | 0.97 |
| $tr_{75\to87}$ | 0.99 | 0.99 | 0.99 | 0.96 | 0.93 | 0.93 | 0.96 | 0.96 |
| $tr_{87\to100}$ | 0.99 | 1.00 | 1.00 | 0.95 | 0.93 | 0.96 | 0.98 | 0.97 |

Table 3: Accuracy matrix for IWMS method on Yearbook dataset.

| Accuracy | $te_{0\to12}$ | $te_{12\to25}$ | $te_{25\to37}$ | $te_{37\to50}$ | $te_{50\to62}$ | $te_{62\to75}$ | $te_{75\to87}$ | $te_{87\to100}$ |
|---|---|---|---|---|---|---|---|---|
| $tr_{0\to12}$ | 0.98 | 0.99 | 0.92 | 0.74 | 0.53 | 0.71 | 0.86 | 0.86 |
| $tr_{12\to25}$ | 0.99 | 0.99 | 0.97 | 0.78 | 0.57 | 0.69 | 0.86 | 0.86 |
| $tr_{25\to37}$ | 0.99 | 0.99 | 0.96 | 0.73 | 0.54 | 0.61 | 0.78 | 0.86 |
| $tr_{37\to50}$ | 0.99 | 1.00 | 0.99 | 0.82 | 0.64 | 0.74 | 0.87 | 0.90 |
| $tr_{50\to62}$ | 0.97 | 0.98 | 0.98 | 0.91 | 0.89 | 0.94 | 0.96 | 0.94 |
| $tr_{62\to75}$ | 0.99 | 0.99 | 0.99 | 0.96 | 0.94 | 0.95 | 0.98 | 0.97 |
| $tr_{75\to87}$ | 0.99 | 1.00 | 0.99 | 0.96 | 0.94 | 0.95 | 0.97 | 0.96 |
| $tr_{87\to100}$ | 0.99 | 1.00 | 1.00 | 0.95 | 0.92 | 0.95 | 0.98 | 0.97 |

## A.18 Online Learning without Memory Rehearsal

The main issue of online learning framework is the lack of information retention mechanisms in traditional methods, which are crucial for addressing the complexities of real world continual learning tasks, such as training a feature extractor that both learns new concepts faster (forward transfer) without losing the capability to perform well on already seen problems (backward transfer).

To highlight that without rehearsing on memory samples the methods suffer significant performance degradation, we implemented the OL algorithm that is mentioned in Section A.12 in the special case in which all the labels (or feedback) arrives in order with a fixed constant delay. We ran new experiments (with identical experimental environment described in the main experimental section, Section 6) on the two largest datasets, CLOC and CGLM, with computational budget respectively, for $d = 10, 50$ and $\mathcal{C} = 2, 8$ respectively. The results show extreme underperformance:

Table 4: Accuracy matrix for Naïve method on CGLM dataset.

| Accuracy | $te_{0\to16}$ | $te_{16\to33}$ | $te_{33\to50}$ | $te_{50\to66}$ | $te_{66\to83}$ | $te_{83\to100}$ |
|---|---|---|---|---|---|---|
| $tr_{0\to16}$ | 0.10 | 0.27 | 0.09 | 0.07 | 0.06 | 0.05 |
| $tr_{16\to33}$ | 0.14 | 0.29 | 0.24 | 0.11 | 0.11 | 0.08 |
| $tr_{33\to50}$ | 0.21 | 0.38 | 0.37 | 0.26 | 0.17 | 0.13 |
| $tr_{50\to66}$ | 0.23 | 0.39 | 0.39 | 0.35 | 0.25 | 0.15 |
| $tr_{66\to83}$ | 0.25 | 0.40 | 0.41 | 0.38 | 0.34 | 0.22 |
| $tr_{83\to100}$ | 0.15 | 0.26 | 0.26 | 0.24 | 0.24 | 0.19 |

Table 5: Accuracy matrix for IWMS method on CGLM dataset.

| Accuracy | $te_{0\to16}$ | $te_{16\to33}$ | $te_{33\to50}$ | $te_{50\to66}$ | $te_{66\to83}$ | $te_{83\to100}$ |
|---|---|---|---|---|---|---|
| $tr_{0\to16}$ | 0.15 | 0.39 | 0.13 | 0.11 | 0.09 | 0.08 |
| $tr_{16\to33}$ | 0.24 | 0.51 | 0.45 | 0.20 | 0.17 | 0.15 |
| $tr_{33\to50}$ | 0.31 | 0.55 | 0.61 | 0.41 | 0.25 | 0.20 |
| $tr_{50\to66}$ | 0.35 | 0.57 | 0.63 | 0.59 | 0.38 | 0.24 |
| $tr_{66\to83}$ | 0.38 | 0.59 | 0.64 | 0.62 | 0.60 | 0.32 |
| $tr_{83\to100}$ | 0.40 | 0.60 | 0.65 | 0.64 | 0.64 | 0.53 |

Table 6: Online Accuracy of Online-Learning (no memory rehearsal) on CLOC

| Time Steps | delay=10 | delay=50 |
|---|---|---|
| 5000 | 0.195 | 0.163 |
| 15000 | 2.142 | 1.354 |
| 25000 | 2.960 | 1.793 |
| 40000 | 3.467 | 2.157 |
| 50000 | 4.202 | 2.451 |
| 60000 | 4.838 | 2.699 |
| 75000 | 5.238 | 2.898 |
| 85000 | 5.632 | 3.076 |
| 95000 | 5.849 | 3.287 |
| 105000 | 6.265 | 3.727 |

Table 7: Online Accuracy of Online-Learning (no memory rehearsal) on CGLM

| Time Steps | delay=10 | delay=50 | delay=100 |
|---|---|---|---|
| 100 | 0.000 | 0.000 | 0.000 |
| 800 | 0.463 | 0.389 | 0.263 |
| 1500 | 0.476 | 0.319 | 0.379 |
| 2200 | 0.531 | 0.242 | 0.257 |
| 2900 | 0.465 | 0.196 | 0.218 |
| 3600 | 0.459 | 0.172 | 0.179 |
| 4300 | 0.390 | 0.188 | 0.187 |
| 5100 | 0.419 | 0.178 | 0.158 |
| 5900 | 0.456 | 0.253 | 0.169 |
| 6600 | 0.504 | 0.313 | 0.175 |

