# OpenReview forum: "Label Delay in Online Continual Learning"
_NeurIPS.cc/2024/Conference — NeurIPS 2024 poster_

### Official Review · Reviewer_UAN4 · 2024-06-20

**Soundness:** 3
**Presentation:** 3
**Contribution:** 2
**Rating:** 5
**Confidence:** 4

**Summary:**

This paper outlines a new continual learning framework to handle label delays in data streams, a situation where labels lag behind data collection. Extensive testing revealed that neither increased computational resources nor advanced techniques like Self-Supervised Learning significantly overcome the challenges posed by label delays. To address this, the authors introduce Importance Weighted Memory Sampling, a method that effectively uses memory samples resembling new unlabeled data to bridge accuracy gaps caused by delays, achieving performance on par with non-delayed scenarios.

**Strengths:**

1. I fully agree with the idea that the assumption of instantaneous annotation in Online Continual Learning (OCL) rarely holds in real-world applications. This makes the following discussions particularly interesting and relevant.
2. The proposed method is simple and easy to understand.

**Weaknesses:**

1. While the discussion on the instantaneous nature of the annotation process is intriguing, I remain curious about the practicality of the proposed Online Continual Learning (OCL) model that incorporates delayed labels. In real-world scenarios, the challenge of matching input data at time step $t$ with labels at time step $t+d$ perfectly may prove more difficult than acquiring a data stream with instantaneous annotations.
2. The proposed setting is not clearly formulized. If the annotator can provide both the image and the corresponding label which makes the memory constraints and proposed label delay quite meaningless, simply train on data stream given by the annotator will be a better choice.
3. Unclear notations. The notation and evaluation criteria in the paper are somewhat unclear. Unless I am mistaken, it seems the model is assessed based on its performance on unlabeled training data. This approach may overlook inherent problems such as forgetting or performance degradation in Online Continual Learning (OCL), since ideally, the model should be evaluated across all previously encountered data distributions. A clearer definition of the OCL model mentioned in the paper is necessary. Additionally, there is notable existing research on online learning methods that handle delayed labels. A comparison with these methods would enrich the discussion and provide a more comprehensive evaluation of the proposed model's effectiveness in the context of OCL.
4. More comparisons with existing OCL methods should be included.
5. The paper does not provide a convincing comparison of its contribution with respect to existing work like[A,B].


[A]: Mesterharm, Chris. "On-line learning with delayed label feedback." International Conference on Algorithmic Learning Theory. Berlin, Heidelberg: Springer Berlin Heidelberg, 2005.

[B]: Gomes, Heitor Murilo, et al. "A survey on semi-supervised learning for delayed partially labelled data streams." ACM Computing Surveys 55.4 (2022): 1-42.

**Questions:**

See the weakness.

---

> ### Author Rebuttal · Authors · 2024-08-07
>
> We thank the reviewer for their efforts reviewing our paper and that the reviewer finds our method to be simple and that they agree on the setup of delayed labels.
>
> **Q1:  the challenge of matching input data at time step 𝑡 with labels at time step 𝑡+𝑑 perfectly may prove more difficult than acquiring a data stream with instantaneous annotations.**
>
> Throughout our work we argue that modeling the delay is essential in OCL settings, and we found that a straightforward experimental model for demonstrating how such delays impact the model performance is the ($x_t, y_{t-d}$) formulation.
>
> Throughout our work, we argue that modeling delay is crucial in OCL settings. We found that a straightforward experimental model, using the (x_t, y_{t-d}) formulation, effectively demonstrates how delays impact model performance. Decoupling the input and label streams could be an interesting generalization, where data and labels arrive asynchronously. We excluded this to maintain focus on our main message. Including this discussion in future work would enhance the manuscript. However, we highlight the numerous experiments varying delay times (as in Figures 2, 3, 4), demonstrating IWMS's superiority over other methods. We do not expect performance rankings to differ significantly with variable delays.
>
> **Q2: The proposed setting is not clearly formalized**
>
> We will review our manuscript to ensure the notations and explanations are as clear as possible and will add clarifications where necessary to prevent misunderstandings. Feedback from other reviewers (bdVn, dj2M, and HwKE) indicated that our presentation was clear, but we understand that improvements can always be made. Please let us know any specific comments the reviewer has in mind.
>
> The Annotator does **not** provide the samples, **only** the labels. The scenario, to which the reviewer refers to, in which we simply wait for the annotator to reveal the labels and ignore the delays, is described as the Naïve case as this is the simplest and in practice the most popular approach.
>
> **Q3: part 1: Unclear notations. The notation and evaluation criteria in the paper are somewhat unclear.**
> We will review our manuscript to ensure the notations and explanations are as clear as possible and will add clarifications where necessary to prevent misunderstandings. Please let us know any specific comments you have in mind.
>
> We appreciate the reviewer's feedback on the evaluation criteria and the suggestion to include more comparisons with existing Online Continual Learning (OCL) methods. However, we would like to clarify both the original OCL formulation ([Cai] in Figure 3c and our manuscript does indeed address the evaluation of forgetting and performance degradation. Specifically, these aspects are thoroughly considered and illustrated in Figure 4 of the main paper, which evaluates the model's performance across all previously encountered data distributions. We apologize if this was not immediately clear and will ensure that the manuscript highlights these evaluations more prominently.
>
> OCL papers often focus less on forgetting and more on online accuracy, i.e., next batch accuracy, because the data stream is chronologically ordered. In this setup, older data (e.g., from 2010) always precedes newer data (e.g., from 2024) and might never reappear. For example, images of Nokia phones are far less common now than those of iPhones. Thus, the goal is to predict future data accurately. Despite this, we still report forgetting and past data accuracy, detailed in the supplementary material.
>
>
> **Q3: part 2: A comparison with these online learning with label delay**
>
> We would like to point the reviewer to the supplementary material, which explains the primary reasons for the incompatibility of traditional online learning approaches with modern OCL settings. Briefly, the main issue is the general lack of information retention mechanisms in traditional methods, which are crucial for addressing the complexities of continual learning tasks.
> Traditional online learning methods often assume immediate access to labels and do not account for the challenges of learning from data streams with delays. In contrast, modern OCL approaches, including our proposed framework, focus on retaining and utilizing past information effectively to mitigate forgetting and adapt to new data distributions.
> We have highlighted these distinctions in our paper and supplementary material to clarify why direct comparisons with traditional methods may not fully capture the unique challenges and contributions of our work. However, we will ensure that these discussions are more clearly articulated in the manuscript.
>
> **Q4: More comparisons with existing OCL**
> Could the reviewer point out which exact method should be included in our comparisons? Please note we used the best performing (under computationally budgeted) OCL as shown recently in [Ghunaim et al.].
>
>
> **Q5: No convincing comparison to [A, B]**
>
> We appreciate the reviewer's suggestion to improve our comparison with prior work. Figures 16 and 17 in the Supplementary Material visually explain the core difference with the framework considered by [B]. In our framework, all labels become available after a fixed duration, making it fully supervised. Gomez et al. do not consider unsupervised and semi-supervised methods for cases where newer data distribution states are accessed using unsupervised techniques, which is our core contribution. We'll emphasize this distinction in the main paper.
>
> Regarding [A], we address the incompatibility issues of such online learning work from 2005 in the related work section, L66. The methods proposed by [A] describe how to wait for labels that may come in a different order than the data points, involving sorting labeled instances or updating on the most recent labels. Our new experiments show that simply rehearsing on the newest supervised data, as [A] suggests, leads to performance collapse (i.e., per chance accuracy).

---

> > ### Author Response · Authors · 2024-08-08
> > **Experimental results on [A]**
> >
> > Briefly, the main issue is the general lack of information retention mechanisms in traditional methods, which are crucial for addressing the complexities of real world continual learning tasks, such as training a feature extractor that both learns new concepts faster (forward transfer) without losing the capability to perform well on already seen problems (backward transfer). To highlight that without rehearsing on memory samples the methods suffer significant performance degradation, we implemented the OL algorithm that is analogous to [A] in the special case in which all the labels (or feedback) arrives in order with a fixed constant delay. We ran new experiments (with identical experimental environment described in the main experimental section, Section 6) on the two largest datasets, CLOC (39M) and CGLM (580K), with computational budget $\mathcal{C}=2, 8$ respectively, for delay parameters $d=10, 50$ and $d=10, 50, 100$ respectively. The results show extreme underperformance:
> >
> > ### Online Accuracy of Online-Learning (no memory rehearsal) on CLOC
> > | Time Steps | delay=10 | delay=50 |
> > | --- | --- | --- |
> > | 5000 | 0.195 | 0.163 |
> > | 15000 | 2.142 | 1.354 |
> > | 25000 | 2.960 | 1.793 |
> > | 40000 | 3.467 | 2.157 |
> > | 50000 | 4.202 | 2.451 |
> > | 60000 | 4.838 | 2.699 |
> > | 75000 | 5.238 | 2.898 |
> > | 85000 | 5.632 | 3.076 |
> > | 95000 | 5.849 | 3.287 |
> > | 105000 | 6.265 | 3.727 |
> >
> >
> >
> > ### Online Accuracy of Online-Learning (no memory rehearsal) on CGLM
> > | Time Steps | delay=10 | delay=50 | delay=100 |
> > | --- | --- | --- | --- |
> > | 100 | 0.000 | 0.000 | 0.000 |
> > | 800 | 0.463 | 0.389 | 0.263 |
> > | 1500 | 0.476 | 0.319 | 0.379 |
> > | 2200 | 0.531 | 0.242 | 0.257 |
> > | 2900 | 0.465 | 0.196 | 0.218 |
> > | 3600 | 0.459 | 0.172 | 0.179 |
> > | 4300 | 0.390 | 0.188 | 0.187 |
> > | 5100 | 0.419 | 0.178 | 0.158 |
> > | 5900 | 0.456 | 0.253 | 0.169 |
> > | 6600 | 0.504 | 0.313 | 0.175 |
> >
> > The results clearly indicate the necessity of memory rehearsal: models on CLOC saturate at <6.5% for delay=10 and <4% for delay=50. In the case of CGLM dataset the performance collapses in all three delay scenarios <1%.

---

> > ### Comment · Reviewer_UAN4 · 2024-08-09
> > **Responses**
> >
> > I appreciate the authors taking the time to respond and help improve my understanding of the manuscript. From my perspective, both online accuracy during incremental learning and overall forgetting performance are crucially important metrics to evaluate continual learning methods, and their relative significance may depend on factors like the specific dataset and intended practical application (e.g. whether domain incremental, class incremental, or both).

---

> ### Author Response · Authors · 2024-08-09
> **Response**
>
> We thank the reviewer for their feedback.
>
> We fully agree with the claim that the significance of these metrics is _highly_ dependent on the exact application. In real world applications often times the final metric is high level, such as "customer satisfaction" or "viewer retention rate" that is difficult if not intractable, therefore these quantitative metrics are just proxies for such higher level objectives. What the research community deems important and relevant can, and indeed does change over time. Our message is simple: "regardless of the choice of the metrics, label delay is a very common problem that needs to be addressed". We truly appreciate the reviewer pointing out the dependency of the relative significance on the specifics of the actual problem.
>
> Regarding the notion of incremental learning, we would like to reiterate on our answer to the reviewer @HwKE, this is a positive signal that we will emphasise in the main manuscript:
> We argue that our experimental setup does not strictly fit the definition of the domain incremental learning, because it does have a changing distribution of the underlying labels as well, while in other benchmarks the labels distribution is static, such as Permuted MNIST, Rotated MNIST [An Empirical Investigation of Catastrophic Forgetting in Gradient-Based Neural Networks by Goodfellow et al.] and Clear [The CLEAR Benchmark: Continual LEArning on Real-World Imagery by Lin et al.]
> An outstandingly clear comparison is provided in Figure 1 of [CLAD: A realistic Continual Learning benchmark for Autonomous Driving by Verwimp et al.] that further refines the definitions of modern continual learning benchmarks.
>
>
> Finally, to ensure that the requirements of the reviewer are met, we would like to point out that we indeed report both metrics in our manuscript:
> - Online accuracy is reported in Figure 2, 3, 4 in the main manuscript and in Figure 5, 9, 11, 12, 13 and 18 in the supplementary material
> - Overall forgetting performance, mentioned by the reviewer is reported as the final score in Figure 19. The curve just gives a richer representation of the performance across all previous tasks. Please find the final scores below:
>
> ### CLOC Dataset
>
> | Delay (d)       | ★ Naïve w/o delay | ◆ Naïve | ✚ S4L | ✖ Pseudo-Label | ▲ IWMS |
> |-----------------|-------------------|---------|-------|----------------|--------|
> | **d=10** (top)  | 6.9 % | 6.0 % | 6.4 % | 6.7 % | 6.6 % |
> | **d=50** (middle) | 6.9 % | 6.2 % | 6.9 % | 6.8 % | 6.9 % |
> | **d=100** (bottom) | 6.9 % | 6.4 % | 6.8 % | 6.7 % | 6.6 % |
>
> ### CGLM Dataset
>
> | Delay (d)       | ★ Naïve w/o delay | ◆ Naïve | ✚ S4L | ✖ Pseudo-Label | ▲ IWMS |
> |-----------------|-------------------|---------|-------|----------------|--------|
> | **d=10** (top)  | 32.2 % | 34.2 % | 28.8 % | 33.1 % | 57.2 % |
> | **d=50** (middle) | 32.2 % | 25.0 % | 33.1 % | 31.8 % | 56.2 % |
> | **d=100** (bottom) | 32.2 % | 36.7 % | 35.1 % | 34.6 % | 56.2 % |
>
> ### FMoW Dataset
>
> | Delay (d)       | ★ Naïve w/o delay | ◆ Naïve | ✚ S4L | ✖ Pseudo-Label | ▲ IWMS |
> |-----------------|-------------------|---------|-------|----------------|--------|
> | **d=10** (top)  | 53.6 % | 56.4 % | 56.8 % | 57.4 % | 55.4 % |
> | **d=50** (middle) | 53.6 % | 56.7 % | 56.6 % | 57.2 % | 55.6 % |
> | **d=100** (bottom) | 53.6 % | 54.4 % | 57.4 % | 56.6 % | 54.3 % |
>
> ### Yearbook Dataset
>
> | Delay (d)       | ★ Naïve w/o delay | ◆ Naïve | ✚ S4L | ✖ Pseudo-Label | ▲ IWMS |
> |-----------------|-------------------|---------|-------|----------------|--------|
> | **d=10** (top)  | 97.2 % | 96.8 % | 97.1 % | 97.3 % | 97.0 % |
> | **d=50** (middle) | 97.2 % | 97.3 % | 94.0 % | 96.4 % | 97.4 % |
> | **d=100** (bottom) | 97.2 % | 94.9 % | 91.0 % | 94.8 % | 95.5 % |

---

> > ### Comment · Reviewer_UAN4 · 2024-08-11
> >
> > I appreciate the authors' thoughtful responses and detailed explanations. It addressed most of my concerns. **I'm raising my score to 5.**

---

> > > ### Author Response · Authors · 2024-08-11
> > >
> > > We thank the reviewer for raising the score and appreciate their recognition of the value of our work.

---

### Official Review · Reviewer_HwKE · 2024-07-07

**Soundness:** 3
**Presentation:** 4
**Contribution:** 3
**Rating:** 6
**Confidence:** 4

**Summary:**

This paper introduces the problem of label delay in Online Continual Learning (OCL), where a model is trained continually on an unlabelled stream where labels are revealed with a fixed delay. The problem becomes learning from semi-supervised data with the objective of quickly adapting to the unlabeled distribution of most recent data. After analyzing the impact of label delay, the authors introduce their method: Importance Weighted Memory Sampling. The main idea is to favor replaying past samples which share similar features representation and labels with current unlabeled samples. Eventually, the authors compare various flagship strategies from different connected domains (Self-Supervsied Semi-Supervised Learning, Test-Time Adaptation and pseudo-labeling) and show superior performances of their approach compared through experiments.

**Strengths:**

- The presentation is very clear and the paper is well-written.
- The motivations of introducing such a realistic problem are clearly defined and I believe the problem of label delay to be interesting to the community.
- Thorough experiments have been conducted, with a specific care given to computation budget, which is often overlooked in OCL
- The idea behind the presented approach is interesting, simple and effective
- The figures are informative and clear

**Weaknesses:**

**Major Weaknesses**

1. My main concern regarding this paper is the evaluation metric. I understand that Online Accuracy is the usual metric in Online Learning. However, in Continual Learning, the usual metric of interest is the Average Accuracy. The problem with Online Accuracy is that only the performances on the last task are considered, meaning that you could have terrible performances on previous tasks. In fact, given that the proposed approach favors samples which are the most similar to the current distribution, I would expect the model to improve on the current task at the cost of worse performances on previous tasks (since the corresponding samples are less likely to be replayed). I have seen the Figure 19 of the appendix regarding backward transfer, but I am unsure that I understand these values correctly. Backward transfer can often be negative, but I might be unfamiliar with those specific datasets. I would like the authors to either:
- Clarify why is Online Accuracy the only metric considered. Is it not important to maintain performances on previous tasks too? Is the lastly observed distribution the only one that is important?
- Show the average accuracy across all tasks in the main draft to ensure that the gain in performances in current task does not come at the cost of lower performances on previous tasks.

2. I wonder why the authors did not compare to existing Continual Learning methods in supervised and semi-supervised cases. Such methods have been cited in appendix (CaSSLE and SCALE). Other methods such as [1,2,3] could also be considered.

3. Why should you have a limited memory size? It seems to me that all data are stored anyway for labeling.

4. The authors did not discuss the potential limitations of their approach.

5. The code is not accessible yet.

**Minor Weaknesses**

6. To my understanding, a similar computational budget does not imply a similar training time. For example, one pass training with Cross-entropy is much faster than one pass with contrastive losses, as they require computing the Gram matrix. it could be enlightening to include training time of various compared methods.

7. This setup is a domain incremental learning setup, I believe this should be stated in the paper.

8. I don't think the memory buffer $M$ is defined in the text.

If the authors can address my main concerns I would happily raise my score.

**Typos**

l191: One -> on ?

l193 Is the memory really $2^{19}$?

[1] Michel, Nicolas, et al. "Contrastive learning for online semi-supervised general continual learning." 2022 IEEE International Conference on Image Processing (ICIP). IEEE, 2022.

[2] Wang, Liyuan, et al. "Ordisco: Effective and efficient usage of incremental unlabeled data for semi-supervised continual learning." Proceedings of the IEEE/CVF Conference on Computer Vision and Pattern Recognition. 2021.

[3] He, Jiangpeng, and Fengqing Zhu. "Unsupervised continual learning via pseudo labels." International Workshop on Continual Semi-Supervised Learning. Cham: Springer International Publishing, 2021.

**Questions:**

See weaknesses.

**Limitations:**

I believe the authors have not addressed the limitations of their work. Please see weaknesses.

---

> ### Author Rebuttal · Authors · 2024-08-07
>
> We thank the reviewer for their recognition of our presentation, the problem we study and the solution we proposed, and appreciate the reviewer for the valuable suggestions. Here are our responses to the reviewer's concerns:
> ## Q1: Evaluation metric
> Using the Online Accuracy metric, in online continual learning, appears to be the common practice in most works, see [Online Continual Learning with Natural Distribution Shifts by Cai et al.] [Real-time evaluation in online continual learning: A new hope by Ghuneim et al] [Computationally budgeted continual learning: What does matter? by Prabhu et al] [Rapid Adaptation in Online Continual Learning by Hammoud et al.] . While evaluation on past tasks has been reported, they were often reported in supplementary as the main focus was the most recent task with OCL. The reason is among the motivation prior works provide, [Cai et al.], is that the top priority for a time-ordered stream is the ability to predict current task and future task (note that evaluation is done before training at each time step). Performance on past tasks is not relevant in case those tasks (which happens to be from the past) might never be presented again (think of images of old laptops or images of devices (for classification) that are no longer popular these days nor will they be in the future.). Thus the majority of works considering OCL, they investigate it on streams that are time ordered (unlike classical offline CL), where performance on next batch accuracy (online accuracy) is the top priority while still reporting the past task accuracies for completion is supplementary.
>
> To reiterate on our results, the reviewer can find the performance on past tasks metric described and evaluated on all settings of the main experiment in Section A.13, Figure 19. The backward transfer in Figure 19 is computed by the average accuracy of the last model on previous data following the evaluation protocol in [Online Continual Learning with Natural Distribution Shifts by Cai et al.]
>
> ## Q2: Comparison to existing Continual Learning methods
> As mentioned in line xx, our work is built on top of previous works of CL with fixed budget [Real-time evaluation in online continual learning: A new hope by Ghuneim et al] [Computationally budgeted continual learning: What does matter? by Prabhu et al] and delayed evaluation [Rapid Adaptation in Online Continual Learning by Hammoud et al.]. As consistently shown in these works, other existing semi-supervised CL methods like CaSSLE and most existing supervised CL methods like regularization based methods are not as effective as ER [On Tiny Episodic Memories in Continual Learning by Chaudhry et al]. [1] is a contrastive learning based method and [3] is a pseudo-labeling based method. We have tried our best to tune these two types of methods and include more advanced variants of them in section 6, although the modified version might be a little bit different from the two suggested ones.
> In conclusion, despite best efforts, since the computational complexity is constrained these methods failr terribly. This is already well documented in several other prior art that conducts these experiments under fixed computation [Rapid Adaptation in Online Continual Learning by Hammoud et al.] [GDumb: A Simple Approach that Questions Our
> Progress in Continual Learning by Prabhu et al.] where more sophisticated methods generally struggle.
>
> Furthermore, prior to this discussion, we tried using the publicly available implementation of [2] and failed to reproduce their results.
>
> ## Q3: Limited memory size
> We agree with the reviewer that we should not limit the buffer size since all data are stored anyway, and this is what we set for the experiments for small datasets, such as FMoW, Wild-Time and CGLM in our experiments section. The current buffer size set is more due to the experiment time and I/O time. We have tried our best to set the buffer size to be $2^{19}$ (line 193), which will be enough to store all the data in a small dataset like Yearbook, FMoW, and CGLM, but not enough for CLOC(39M, Supplementary A1). We further have buffer size analysis in supplementary A14, showing that our methods is robust to different buffer sizes and the performance is not relied on large buffer size.
>
> ## Q4: Potential limitations and code accessibility
> We agree with the reviewer that we should discuss the potential limitations of our approach. We will add this in the final version of the paper. We will also make our code accessible after the paper is accepted.
>
> ## Q5: Training time
> We agree with the reviewer that the training time can be influenced by various factors. Our current computation budget is based on the number of forward-backward passes, which is generally a good proxy for time and it has been widely used in CL literature[8,9].
> While the training time metric could be sensitive to multiple other factors, e.g. code optimization, hardware, data I/O speed, and implementation.
>
> Here we add the training time for our method, ER, constrastive learning-based method, pseudo-labeling based method, and TTA methods  with the same number of forward-backward pass in Table 1.
> Most of the experiments are using a single A100 GPU with 12 CPU. The training time is measured in hours. This table shows that the training time is could be entirely different due to various other factors.
>
>
> |  | CLOC | CGLM | FMoW | Yearbook |
> |---|:---:|:---:|:---:|:---:|
> | Naive | 52 | 3.6 | 2.3 | 0.2 |
> | ReSSL | 67.3 | 3 | 4 | 0.25 |
> | CoTTA | 39 | 5 | 2.5 | 0.2 |
> | Pseudo Labeling | 111.3$^1$ | 4.6 | 2.5 | 0.15 |
> | IWMs | 61 | 3.6 | 3.5 | 0.2 |
> $^1$ the CPU allocation was 6

---

> > ### Author Response · Authors · 2024-08-07
> > **Additional comments**
> >
> > > The code is not accessible yet.
> >
> > Thank you for requesting the code, we believe that publishing code will help improve transparency reproducibility. To that end, following the NeurIPS 2024 Authors' guideline, we have submitted an anonymised link to the following two codebases:
> > - an online interactive demo written in JavaScript, using the webcam as the input data stream in which we visualise how our experimental framework defines label delay and how IWMS selects samples from the buffer
> >
> > - the original experimental framework that was used for the entirety of the project. We were logging every experiment in Weights and Biases for reproducibility, however we could not transfer the project to an anonymous user to share it. Nevertheless, upon acceptance, we will make the project, with all our findings publicly accessible.
> >
> >
> > > I don't think the memory buffer $M$ is defined in the text.
> >
> > We thank the reviewer for pointing this detail out. Our implementation of the memory is identical to [Online Continual Learning with Natural Distribution Shifts by Cai et al.]. We will update this in the paper alongside with the mentioned typos.
> >
> > > This setup is a domain incremental learning setup, I believe this should be stated in the paper.
> >
> > We argue that our experimental setup does not strictly fit the definition of the domain incremental learning, because it does have a changing distribution of the underlying labels as well, while in other benchmarks the labels distribution is static, such as Permuted MNIST, Rotated MNIST [An Empirical Investigation of Catastrophic Forgetting in Gradient-Based Neural Networks by Goodfellow et al.] and Clear [The CLEAR Benchmark: Continual LEArning on Real-World Imagery by Lin et al.]
> >
> > An outstandingly clear comparison is provided in Figure 1 of [CLAD: A realistic Continual Learning benchmark for Autonomous Driving by Verwimp et al.] that further refines the definitions of modern continual learning benchmarks.

---

> ### Comment · Reviewer_HwKE · 2024-08-11
>
> I thank the authors for taking the time to address my concerns. I have read their rebuttal carefully.
>
> **Evaluation Metric**
>
> I get the authors point, although I respectfully disagree. The Online Accuracy is *not* the common practice in most work in OCL. See [1,2,3,4,5,6,7,8] for examples. Now, I understand that the metric is indeed very dependent on the specific application and that in your case of study you focus on Online Accuracy. While this makes sense, I believe this should be clarified in the manuscript, since I do not think that Online Accuracy is obviously the most important metric.
>
> Regarding Figure 19, I do not think this is the standard definition of Backward Transfer, or at least it differs from the definition of [1]. In any case, if my understanding is correct, IWMS indeed improves Online Accuracy at the cost of some marginal performance drop on previous tasks for specific cases (FMoW dataset). This does not in any case diminish this work's quality, although I believe that this potential drawback of the approach could be more clearly stated in the manuscript, in a limitation section for example.
>
> **Comparison to existing OCL methods**
>
> While your work is based on previous studies, their findings regard the *fully supervised* scenario. I still believe that including some existing work on Semi-Supervised OCL in your comparison would improve the manuscript.
>
> **About the setup**
>
> Thank you for clarifying. I understand the setup more clearly now. However, I am not sure to understand, if new classes appear in the stream, but are unlabeled yet, how do you predict them? To my understanding, you would not predict them before they are labeled.
>
> **Training time**
>
> Thank you for sharing the training time, I believe including it in the appendix would improve the paper.
>
> **Code**
>
> Thank you for sharing the code with a live demo.
>
> [1] Mai, Zheda, et al. "Online continual learning in image classification: An empirical survey." Neurocomputing 469 (2022): 28-51.
>
> [2] Guo, Yiduo, Bing Liu, and Dongyan Zhao. "Online continual learning through mutual information maximization." International conference on machine learning. PMLR, 2022.
>
> [3] Koh, Hyunseo, et al. "Online boundary-free continual learning by scheduled data prior." The Eleventh International Conference on Learning Representations. 2023.
>
> [4] Wang, Maorong, et al. "Improving Plasticity in Online Continual Learning via Collaborative Learning." Proceedings of the IEEE/CVF Conference on Computer Vision and Pattern Recognition. 2024.
>
> [5] Wei, Yujie, et al. "Online prototype learning for online continual learning." Proceedings of the IEEE/CVF International Conference on Computer Vision. 2023.
>
> [6] Gu, Yanan, et al. "Not just selection, but exploration: Online class-incremental continual learning via dual view consistency." Proceedings of the IEEE/CVF Conference on Computer Vision and Pattern Recognition. 2022.
>
> [7] Buzzega, Pietro, et al. "Dark experience for general continual learning: a strong, simple baseline." Advances in neural information processing systems 33 (2020): 15920-15930.
>
> [8] Prabhu, Ameya, Philip HS Torr, and Puneet K. Dokania. "Gdumb: A simple approach that questions our progress in continual learning." Computer Vision–ECCV 2020: 16th European Conference, Glasgow, UK, August 23–28, 2020, Proceedings, Part II 16. Springer International Publishing, 2020.

---

> ### Author Response · Authors · 2024-08-14
>
> We thank the reviewer for engaging in this thoughtful conversation and bringing our attention towards richer representations of performance in OCL settings.
>
> We would like to respond to the raised points in order:
>
> ## First point
> > The Online Accuracy is not the common practice in most work in OCL. See [1,2,3,4,5,6,7,8] for examples.
>
> ### Comment on [1]
>
> In the referenced paper [1], under the "Evaluation Metrics" Average Accuracy is defined as follows:
>
> $$\mathrm{Average Accuracy}(A_i)=\frac{1}{i}\sum_{j=1}^ia_{i,j}$$
>
> which is closely related to Online Accuracy, but we appreciate the fact that in such scenario performance on past iterations are re-evaluated in every step. In our setting evaluating and reporting this metric would be infeasible, as the number of steps on the CLOC, CGLM and FMoW dataset is *3-to-4* orders of magnitude larger than the examples provided in the survey (the maximum number of steps in [1] is 20, for comparison in our experiments the maximum number of "tasks" is 296 119).
>
> > Regarding Figure 19, I do not think this is the standard definition of Backward Transfer, or at least it differs from the definition of [1].
>
> We truly appreciate the reviewer pointing out the difference in the definition.
> We recognise that the backward and forward transfer can, and should be reported (as defined in [1]) at given points during the training.
> To this end we have re-run the Naïve and IWMS experiments with $d=50$ on the two datasets where IWMS was performing the _best_ and the _worst_ to provide a full comparison against the baseline.
> We simplified the table representation by splitting the validation data into 100 equal sized ranges along the time axis, such that the ranges would correspond to the training data range:

---

> > ### Author Response · Authors · 2024-08-14
> >
> > #### Accuracy matrix of Naïve on Yearbook @ $d=50$
> > | Accuracy | $te_{0 \rightarrow 12 }$ | $te_{12 \rightarrow 25 }$ | $te_{25 \rightarrow 37 }$ | $te_{37 \rightarrow 50 }$ | $te_{50 \rightarrow 62 }$ | $te_{62 \rightarrow 75 }$ | $te_{75 \rightarrow 87 }$ | $te_{87 \rightarrow 100 }$ |
> > | -------- | ------------------------ | ------------------------- | ------------------------- | ------------------------- | ------------------------- | ------------------------- | ------------------------- | -------------------------- |
> > | $tr_{ 0 \rightarrow 12 }$ | 0.99 | 0.98 | 0.94 | 0.76 | 0.56 | 0.70 | 0.89 | 0.88 |
> > | $tr_{ 12 \rightarrow 25 }$ | 0.98 | 0.99 | 0.97 | 0.79 | 0.58 | 0.72 | 0.88 | 0.86 |
> > | $tr_{ 25 \rightarrow 37 }$ | 0.99 | 0.99 | 0.96 | 0.75 | 0.55 | 0.59 | 0.77 | 0.86 |
> > | $tr_{ 37 \rightarrow 50 }$ | 0.99 | 1.00 | 0.99 | 0.83 | 0.65 | 0.74 | 0.87 | 0.90 |
> > | $tr_{ 50 \rightarrow 62 }$ | 0.94 | 0.96 | 0.96 | 0.88 | 0.88 | 0.94 | 0.96 | 0.94 |
> > | $tr_{ 62 \rightarrow 75 }$ | 0.97 | 0.99 | 0.99 | 0.96 | 0.93 | 0.93 | 0.97 | 0.97 |
> > | $tr_{ 75 \rightarrow 87 }$ | 0.99 | 0.99 | 0.99 | 0.96 | 0.93 | 0.93 | 0.96 | 0.96 |
> > | $tr_{ 87 \rightarrow 100 }$ | 0.99 | 0.99 | 1.00 | 0.95 | 0.93 | 0.96 | 0.98 | 0.97 |
> >
> >
> > #### Accuracy matrix of IWMS on Yearbook @ $d=50$
> > | Accuracy | $te_{0 \rightarrow 12 }$ | $te_{12 \rightarrow 25 }$ | $te_{25 \rightarrow 37 }$ | $te_{37 \rightarrow 50 }$ | $te_{50 \rightarrow 62 }$ | $te_{62 \rightarrow 75 }$ | $te_{75 \rightarrow 87 }$ | $te_{87 \rightarrow 100 }$ |
> > | -------- | ------------------------ | ------------------------- | ------------------------- | ------------------------- | ------------------------- | ------------------------- | ------------------------- | -------------------------- |
> > | $tr_{ 0 \rightarrow 12 }$ | 0.98 | 0.99 | 0.92 | 0.74 | 0.53 | 0.71 | 0.86 | 0.86 |
> > | $tr_{ 12 \rightarrow 25 }$ | 0.99 | 0.99 | 0.97 | 0.78 | 0.57 | 0.69 | 0.86 | 0.86 |
> > | $tr_{ 25 \rightarrow 37 }$ | 0.99 | 0.99 | 0.96 | 0.73 | 0.54 | 0.61 | 0.78 | 0.86 |
> > | $tr_{ 37 \rightarrow 50 }$ | 0.99 | 1.00 | 0.99 | 0.82 | 0.64 | 0.74 | 0.87 | 0.90 |
> > | $tr_{ 50 \rightarrow 62 }$ | 0.97 | 0.98 | 0.98 | 0.91 | 0.89 | 0.94 | 0.96 | 0.94 |
> > | $tr_{ 62 \rightarrow 75 }$ | 0.99 | 0.99 | 0.99 | 0.96 | 0.94 | 0.95 | 0.98 | 0.97 |
> > | $tr_{ 75 \rightarrow 87 }$ | 0.99 | 1.00 | 0.99 | 0.96 | 0.94 | 0.95 | 0.97 | 0.96 |
> > | $tr_{ 87 \rightarrow 100 }$ | 0.99 | 1.00 | 1.00 | 0.95 | 0.92 | 0.95 | 0.98 | 0.97 |
> >
> > #### Accuracy matrix of Naïve on CGLM @ $d=50$
> > | Accuracy | $te_{0 \rightarrow 16 }$ | $te_{16 \rightarrow 33 }$ | $te_{33 \rightarrow 50 }$ | $te_{50 \rightarrow 66 }$ | $te_{66 \rightarrow 83 }$ | $te_{83 \rightarrow 100 }$ |
> > | -------- | ------------------------ | ------------------------- | ------------------------- | ------------------------- | ------------------------- | -------------------------- |
> > | $tr_{ 0 \rightarrow 16 }$ | 0.10 | 0.27 | 0.09 | 0.07 | 0.06 | 0.05 |
> > | $tr_{ 16 \rightarrow 33 }$ | 0.14 | 0.29 | 0.24 | 0.11 | 0.11 | 0.08 |
> > | $tr_{ 33 \rightarrow 50 }$ | 0.21 | 0.38 | 0.37 | 0.26 | 0.17 | 0.13 |
> > | $tr_{ 50 \rightarrow 66 }$ | 0.23 | 0.39 | 0.39 | 0.35 | 0.25 | 0.15 |
> > | $tr_{ 66 \rightarrow 83 }$ | 0.25 | 0.40 | 0.41 | 0.38 | 0.34 | 0.22 |
> > | $tr_{ 83 \rightarrow 100 }$ | 0.15 | 0.26 | 0.26 | 0.24 | 0.24 | 0.19 |
> >
> > #### Accuracy matrix of IWMS on CGLM @ $d=50$
> > | Accuracy | $te_{0 \rightarrow 16 }$ | $te_{16 \rightarrow 33 }$ | $te_{33 \rightarrow 50 }$ | $te_{50 \rightarrow 66 }$ | $te_{66 \rightarrow 83 }$ | $te_{83 \rightarrow 100 }$ |
> > | -------- | ------------------------ | ------------------------- | ------------------------- | ------------------------- | ------------------------- | -------------------------- |
> > | $tr_{ 0 \rightarrow 16 }$ | 0.15 | 0.39 | 0.13 | 0.11 | 0.09 | 0.08 |
> > | $tr_{ 16 \rightarrow 33 }$ | 0.24 | 0.51 | 0.45 | 0.20 | 0.17 | 0.15 |
> > | $tr_{ 33 \rightarrow 50 }$ | 0.31 | 0.55 | 0.61 | 0.41 | 0.25 | 0.20 |
> > | $tr_{ 50 \rightarrow 66 }$ | 0.35 | 0.57 | 0.63 | 0.59 | 0.38 | 0.24 |
> > | $tr_{ 66 \rightarrow 83 }$ | 0.38 | 0.59 | 0.64 | 0.62 | 0.60 | 0.32 |
> > | $tr_{ 83 \rightarrow 100 }$ | 0.40 | 0.60 | 0.65 | 0.64 | 0.64 | 0.53 |

---

> ### Author Response · Authors · 2024-08-14
>
> >  In any case, if my understanding is correct, IWMS indeed improves Online Accuracy at the cost of some marginal performance drop on previous tasks for specific cases (FMoW dataset).
>
> It is correct, thank you for highlighting this.
> We will address the limitations on backward transfer on FMoW.
>
> ### Comment on [2, 5, 6, 7]
>
> We would like to point out that Guo et al. details the evaluation metric in 6.1 as follows:
>
> > We first learn from the data stream of all tasks for
> each dataset, and then test the final model using the test data
> of all tasks. We report the average accuracy of all tasks from
> 15 random runs for each dataset
>
> We would like to argue that this metric is taking an excessively strong measure to remove noise from the metric. In our experiments, we experienced that rerunning the same training with different seed results in negligible (less than 0.01%) differences in the results. Running the experiments 15 times to evaluate the metric of [2] is infeasible for us.
>
> This similarly hold for [5] as well, since they report the Average Accuracy and Average Forgetting across 15 runs.
>
> Furthermore, although the main manuscript of [6] does not provide the detail about re-running the Average Accuracy, the corresponding code is set by default to 15 re-runs with different seeds. (follow this URL for reference:
> https://github.com/YananGu/DVC/blob/6f12984d10a4a1c4609f221b939f93d94fc8258e/general_main.py#L29 )
>
> In [7], the number of random initialization is dropped to 10, otherwise they report the Average Accuracy as well.
>
> ### Comment on [3]
> We would like to point out that [3] introduces their own metric: Knowledge Loss/Gain Ratio [3] claiming that the metrics used by [2, 5, 6] are relying on the notion of task boundaries therefore they define a new objective that is "appropriate for periodic data distribution".
> In our paper we cannot make such assumptions about periodicity.
>
>
> ### Comment on [4]
> The accuracy metric proposed by [4], Learning Accuracy (LA) using Model Plasticity is formally defined for the $j$-th task as:
> $$
>  l_j = a_j^j
> $$
> where $a_j^i$ is the accuracy evaluated on the test set of task
> $j$ after training the network from task 1 to task $i$.
> We would like to argue that this metric is similar to the _Online Accuracy_ metric apart from the notion that here the test samples are drawn from a different distribution, whereas the Online Accuracy is evaluated on the $j$-th batch of data before it is used for training.
> If we assume that both the test and the training batch is drawn from the same distribution at time-step $j$, the two metrics are arguably the same.
> (Please note that the training is only done on the batch after the evaluation in the case of Online Accuracy.)
>
> ### Comment on [8]
> The paper _"GDumb: A simple approach that questions our progress in continual learning"_ highlights the weakness of the then-standard metrics in Continual Learning such as Average Accuracy and Accuracy at end. While the paper aimed to steer the CL research community towards higher standards, these metrics stayed popular.
>
> In fact, [Online Continual Learning with Natural Distribution Shifts by Zai et al.] uses GDumb's argument to propose the Online Accuracy metric that recently increasing number of work adopted.
>
> ### Resolution
> > Now, I understand that the metric is indeed very dependent on the specific application and that in your case of study you focus on Online Accuracy. While this makes sense, I believe this should be clarified in the manuscript, since I do not think that Online Accuracy is obviously the most important metric.
>
> We recognize and fully agree with the reviewer's criticism that our choice of metric needs more justification, as we present it as the "go-to" metric in our narrative.
> We will definitely address these points in the updated manuscripts more carefully, if the reviewer agrees.
>
>
> ## Second point
> > While your work is based on previous studies, their findings regard the fully supervised scenario. I still believe that including some existing work on Semi-Supervised OCL in your comparison would improve the manuscript.
>
> Due to time limitations we could not fully reimplement [Contrastive learning for online semi-supervised general continual learning. by Michel et al.] to report scores in the rebuttal period, but we are happy to include them in the final manuscript.
>
> For reference, while we think the comparison would indeed interesting, we remain skeptical about contrastive methods outperforming IWMS, or even the Naïve baseline under our experiments' computational constraints, as we discuss the results of 6 different contrastive methods in detail in the Supplementary Material A.8.

---

> > ### Author Response · Authors · 2024-08-14
> >
> > ## Third point
> > > However, I am not sure to understand, if new classes appear in the stream, but are unlabeled yet, how do you predict them? To my understanding, you would not predict them before they are labeled.
> >
> > In our setup the model **has** to make a prediction at every time-step, even if an unseen category is presented for the first time (which is a realistic scenario). In such cases it is indeed theoretically impossible to correctly predict the class (apart from the accidental random-chance correct guesses), and the model is not able to perform well on the new class until the labels arrive. In our setup we simply penalize the model, regardless whether a class was seen or not. This can be also verified in the live demo (press the "`" key to open the Online and the Current Accuracy curve panel).
> >
> > ## Final note
> > We would like to thank the reviewer for their insightful comments and opening up the discussion.
> > We hope that we have addressed all concerns in detail.

---

> ### Comment · Reviewer_HwKE · 2024-08-14
> **Thank you for the detailed responses**
>
> I would like to thank the authors for taking the time to address all my points and for their detailed response.
>
> I agree that Average Accuracy is not the *only* metric considered in the paper I referenced, but it is still the most commonly used one for comparison. I guess we could argue on which is the most important one depending on the problem at hand, and again I understand that Online Accuracy might make more sense in your context.
>
> *Overall conclusion*
>
> The authors have addressed my concerns and I will **increase my score to 6**.

---

### Official Review · Reviewer_dj2M · 2024-07-10

**Soundness:** 2
**Presentation:** 3
**Contribution:** 2
**Rating:** 5
**Confidence:** 4

**Summary:**

This manuscript delineates a novel method, termed as IWMS, which is devised to address the problem of label delay in online continual learning, where new data may not be labeled due to slow and costly annotation processes. The IWMS exhibits an innovative usage of fine-grained Gaussian Mixture prototypes,  along with mutual information optimizing that endow the proposed method with competitive performance for unsupervised class incremental discovery.

**Strengths:**

S1. The paper's significance is underscored by its empirical evaluation. The evaluation is comprehensive, with comparisons to different baselines and ablation study providing a compelling demonstration of the strength of this work.

S2. The proposed IWMS method is simple and easy to implement. The proposed memory sampling has the potential to inspire future research in this domain.

**Weaknesses:**

W1. Although this paper presents a simple method, further technical insights regarding the implementation and each module within the IWMS method would be beneficial. Further efforts regarding technical innovation and methodological novelty would also be beneficial.

W2. The manuscript could delve deeper into the memory buffer (e.g., how to construct and update it, and the additional complexity by it), a factor which could be pivotal for fully understanding the proposed model.

W3. It would be better to analyze and discuss the difference between online learning, continual learning, and online continual learning, and further clarify the topic of this work.

**Questions:**

See weaknesses above.

**Limitations:**

Addressed.

---

> ### Author Rebuttal · Authors · 2024-08-07
>
> We thank the reviewer's recognition for our extensive experiments and the simplicity of our proposed method and the value suggestions. The following are our responses to the reviewer's concerns:
>
> **Q1: Need further explanation regarding the implementation and each module within the IWMS method and technical innovation.**
>
> Our implementation of IWMs is closely related to the experimental setup in Section 5. It is built upon the best performing method, ER, and we further improve it by sampling the buffer samples that are more similar to the unlabeled data. All the implementation details and "technical innovation" are detailed in full in Section 1, 3 and 4, in Figure 1 and in Algorithm 1 and 2. Line 149 - provides "technical insights regarding the implementation and each module within the IWMS method". Line 154 and 163 - "delves deeper into the memory buffer". Section A.12, Line 756 - Analyses the difference between online learning continual learning and online continual learning.
>
> As we point it out in the main paper in Section 4, L137, the novelty lies in how to learn the unlabeled distribution.  Our experiments in Section 6 demonstrate that existing literature, which focus on operating unlabeled features to learn the most recent distribution, are often neither effective nor efficient. Our approach avoids unnecessary computation on the most recent unlabeled distribution, even if it closely matches the evaluation distribution. Instead, we sample previously labeled data similar to the current unlabeled data. This allows us to construct a pseudo-distribution close to the most recent distribution and conduct purely supervised learning on the pseudo-distribution.
>
> **Q2: Need further explanation regarding the memory buffer.**
>
> This is all detailed in Section 4, Line 154, but we will reiterate it here for completion.
> At time step $t$, we keep a memory buffer to store both the raw data and the features of the labeled data from time step $t'$($t'$< $t$) computed in $t'$. We sample the data according to the similarity between the features from the current unlabeled data $x^t$ and the features in the buffer. We then take the buffer data whose features are most similar to x^t to construct a new labeled batch. This batch is our buffer batch.
>
> **Q3: Need to analyze and discuss the difference between online learning, continual learning, and online continual learning, and further clarify the topic of this work.**
>
> We already had a Section 2 and A.12  in the paper discussing this in full depth. In fact, the title of the section is "Online Learning vs Online Continual Learning" and "Considering catastrophic forgetting".
>
> We provide further explanation below:
> Online Learning and Online Continual Learning both involve learning from data arriving sequentially, but Online Learning typically deals with single-task streams, often assumed to be from an i.i.d. distribution. In contrast, Online Continual Learning is more concerned with non-stationary streams that undergo frequent changes in distribution. Continual Learning, on the other hand, is a broader concept that encompasses any type of learning process where the data is revealed sequentially and the distribution of the data may change over time. Recent discussion in continual learning has focused on the scenario where the computation is fixed for fair comparison.
>
> Our work mainly focuses on the online continual learning scenario, where the distribution of the data changes over time and the data is revealed batch-wise. As motivated in line 26, we further consider the label delay problem, where the labels of the data are not available immediately, and most recent data distribution is unlabeled. To this end, we propose solutions from various literatures and propose our simple yet effective method, IWMs, to address the label delay problem in online continual learning.

---

> > ### Author Response · Authors · 2024-08-14
> >
> > We would like to ask if we managed to address the concerns of the reviewer?

---

### Official Review · Reviewer_bdVn · 2024-07-14

**Soundness:** 3
**Presentation:** 3
**Contribution:** 2
**Rating:** 6
**Confidence:** 4

**Summary:**

The paper addresses the problem of label delay in online continual learning. The proposed framework explicitly models this delay, revealing unlabeled data from the current time step and labels delayed by a specific number of steps. Extensive experiments demonstrate that increasing computational resources alone is insufficient to overcome performance declines caused by significant label delays. The authors introduce Importance Weighted Memory Sampling, a robust method that prioritizes memory samples similar to the newest unlabeled data, improve the performance under label delays scenario.

**Strengths:**

-   The paper is well-written and easy to follow.

-   The proposed method is comprehensive and reasonable. In Section 4, the IWMS addresses the critical issue of outdated feature updates.

-   The experiments conducted in the paper are thorough and convincing, covering a variety of datasets and including ablation studies, sensitivity analyses, and more.

**Weaknesses:**

1. The discussion of related work on the label delay problem is not comprehensive. The authors should consider additional relevant studies in the field of online learning with label delay, such as:

    - Heliou et al. "Gradient-free Online Learning in Continuous Games with Delayed Rewards." ICML 2020.
    - Wan et al. "Online Strongly Convex Optimization with Unknown Delays." Machine Learning 2022.

    Additionally, recent works on "asynchronous labels" (which is a general version of label delay, in which both the feature and label can delay) should be better considered:

    - Zheng et al. "Asynchronous Stochastic Gradient Descent with Delay Compensation." ICML 2017.
    - Qian et al. "Learning with Asynchronous Labels." TKDD 2024.

2. The paper considers only the simple case of a fixed delay time step $d$, which may not be practical for real-world scenarios. It would be more beneficial if the authors considered a more general case where the delay time step is variable.

3. Although previous research on online learning with delayed feedback focuses on the online learning scenario rather than continual learning, it would be more comprehensive if the authors compared their method with these online learning approaches.

**Questions:**

See Weakness above.

**Limitations:**

See Weakness above.

---

> ### Author Rebuttal · Authors · 2024-08-07
>
> We appreciate the reviewers' positive comments on our method, experiments, and paper presentation. We address the above concerns here.
>
> **Q1: The discussion of related work on the label delay problem is not comprehensive.**
>
> We thank the reviewers for pointing out the missing references. We will include the suggested references in the final version of the paper, covering both the label delay and asynchronous labels scenarios.
>
> **Q2: The paper considers only the simple case of a fixed delay time step d.**
>
> We acknowledge that this is an excellent suggestion and plan to add further ablations on varying delay time steps during the rebuttal period. However, we would like to point out to the reviewer the numerous experiments we have conducted where we vary $d$  across all experiments  (as in Figure 2,3,4), which already demonstrate the superiority of IWMS over other methods. We do not generally expect the performance ranking to differ significantly with variable delay times.
>
> **Q3: Comparison to the online learning approaches**
>
> We have a thorough discussion on this point in the supplementary A12. Furthermore, We appreciate the reviewer’s suggestion to compare our work with existing research on online learning methods that handle delayed labels. We would like to point the reviewer to the supplementary material A12 (page 23, L756), which explains the primary reasons for the incompatibility of traditional online learning approaches with modern OCL settings. Briefly, the main issue is the general lack of information retention mechanisms in traditional methods, which are crucial for addressing the complexities of real world continual learning tasks, such as training a feature extractor that both learns new concepts faster (forward transfer) without losing the capability to perform well on already seen problems (backward transfer). To highlight that without rehearsing on memory samples the methods suffer significant performance degradation, we implemented the OL algorithm that is analogous to the mentioned papers in the special case in which all the labels (or feedback) arrives in order with a fixed constant $d$ delay.
> We ran new experiments (with identical experimental environment described in the main experimental section, Section 6) on the two largest datasets, CLOC (39M) and CGLM (580K), with computational budget $\mathcal{C}=2,8$ respectively, for $d=10,50$ and $d=10, 50, 100$ respectively. The results show extreme underperformance:
>
> [Online Accuracy of Online-Learning (no memory rehearsal) on CLOC]
>
> | Time Steps | delay=10 | delay=50 |
> | --- | --- | --- |
> | 5000 | 0.195 | 0.163 |
> | 15000 | 2.142 | 1.354 |
> | 25000 | 2.960 | 1.793 |
> | 40000 | 3.467 | 2.157 |
> | 50000 | 4.202 | 2.451 |
> | 60000 | 4.838 | 2.699 |
> | 75000 | 5.238 | 2.898 |
> | 85000 | 5.632 | 3.076 |
> | 95000 | 5.849 | 3.287 |
> | 105000 | 6.265 | 3.727 |
>
> [Online Accuracy of Online-Learning (no memory rehearsal) on CGLM]
>
> | Time Steps | delay=10 | delay=50 | delay=100 |
> | --- | --- | --- | --- |
> | 100 | 0.000 | 0.000 | 0.000 |
> | 800 | 0.463 | 0.389 | 0.263 |
> | 1500 | 0.476 | 0.319 | 0.379 |
> | 2200 | 0.531 | 0.242 | 0.257 |
> | 2900 | 0.465 | 0.196 | 0.218 |
> | 3600 | 0.459 | 0.172 | 0.179 |
> | 4300 | 0.390 | 0.188 | 0.187 |
> | 5100 | 0.419 | 0.178 | 0.158 |
> | 5900 | 0.456 | 0.253 | 0.169 |
> | 6600 | 0.504 | 0.313 | 0.175 |
>
> The results clearly indicate the necessity of memory rehearsal:
> models on CLOC saturate at <6.5% for delay=10 and <4% for delay=50. In the case of CGLM dataset the performance collapses in all three delay scenarios <1%.

---

> > ### Comment · Reviewer_bdVn · 2024-08-12
> >
> > The authors' responses have addressed my concerns, so I have decided to raise my score. I would be happy to see a more comprehensive discussion of related works in the revised version!

---

> > > ### Author Response · Authors · 2024-08-14
> > >
> > > We thank the reviewer for recognizing the value of our proposal.
> > >
> > > We will provide an in depth comparison to the papers discussed in the rebuttal period in the final version.

---

### Author Response · Authors · 2024-08-09
**Code for Interactive Javascript visualization and experimental framework**

Several authors requested access to the code and we believe that publishing code will help improve transparency reproducibility.
To that end, following the NeurIPS 2024 Authors' guideline, we have submitted an anonymised link to the following two codebases:

- an online interactive demo written in JavaScript, using the webcam as the input data stream in which we visualise how our experimental framework defines label delay and how IWMS selects samples from the buffer

- the original experimental framework that was used for the entirety of the project. We were logging every experiment in Weights and Biases for reproducibility, however we could not transfer the project to an anonymous user to share it. Nevertheless, upon acceptance, we will make the project, with all our findings publicly accessible.

The code can be accessed for 6 days via the following anonymous link (with names and other personal references removed from the comments): https://filebin.net/k0mx9ru6607uc4g9

---

### Decision · Program_Chairs · 2024-09-25

**Decision:**

Accept (poster)

**Comment:**

This paper introduces the concept of label delay (well studied in online learning) into the online continual learning setting. The authors first show that a naive method (ignoring the unlabelled data) degrades rapidly as the delay increases (as expected). They then show that existing methods that make use of unsupervised data, such as test-time adaptation (TTA) don’t solve the problem. They then introduce importance weighted memory sampling (IWMS) as a simple method that is claimed to outperform other methods.

The reviewers cited various strengths and weaknesses leading to scores (post author responses) of 6, 5, 6, and 5:
Strengths:
- [bdVn, HwKE] well written / easy to follow
- [bdVn, dj2m, UAN4] IWMS seems like a reasonable solution
- [bdVn, dj2m, HwKE] comprehensive experiments
Weaknesses:
- [bdVn] missing related work
- [bdVn, UAN4] fixed delay assumption may be unreasonable
- [bdVn, HwKE, UAN4] missing comparisons - e.g. to label delay in online learning or other semi supervised or online continual learning methods
- [HwKE] choice of online accuracy as a metric
- [HwKE] discussion of limitations
- [HwKE] availability of code

There was active discussion, and many of these points were resolved satisfactorily. One sticking point was around the choice of metric, but on its own I don’t feel this is reason to reject the paper. Regarding the code request, the authors did put (time limited) code up. I had a look through myself, but found it hard to evaluate without executing as there was a lack of documentation. Some readme files were later added, but I didn’t get a chance to read these. At least the “code will be made available” box can be checked.
The authors also point out that the review from dj2m appears to be LLM generated, and in fact mentions methods that don’t appear in the paper. This review was also the least thorough of the 4, so I would give it less weight in the overall evaluation anyway.

Overall the work appears to be a useful contribution to the field, and is of a high enough standard for the conference. As such I am recommending that the paper be accepted.